# NUP98 and RAE1 sustain progenitor function through HDAC-dependent chromatin targeting to escape from nucleolar localization

Amy E. Neely[1], Laura A. Blumensaadt[1], Patric J. Ho[1], Sarah M. Lloyd [1], Junghun Kweon[1], Ziyou Ren[2] & Xiaomin Bao [1,2,3,4✉]

Self-renewing somatic tissues rely on progenitors to support the continuous tissue regeneration. The gene regulatory network maintaining progenitor function remains incompletely understood. Here we show that NUP98 and RAE1 are highly expressed in epidermal progenitors, forming a separate complex in the nucleoplasm. Reduction of NUP98 or RAE1 abolishes progenitors' regenerative capacity, inhibiting proliferation and inducing premature terminal differentiation. Mechanistically, NUP98 binds on chromatin near the transcription start sites of key epigenetic regulators (such as *DNMT1*, *UHRF1* and *EZH2*) and sustains their expression in progenitors. NUP98's chromatin binding sites are co-occupied by HDAC1. HDAC inhibition diminishes NUP98's chromatin binding and dysregulates NUP98 and RAE1's target gene expression. Interestingly, HDAC inhibition further induces NUP98 and RAE1 to localize interdependently to the nucleolus. These findings identified a pathway in progenitor maintenance, where HDAC activity directs the high levels of NUP98 and RAE1 to directly control key epigenetic regulators, escaping from nucleolar aggregation.

[1] Department of Molecular Biosciences, Northwestern University, Evanston, IL, USA. [2] Department of Dermatology, Northwestern University, Chicago, IL, USA. [3] Simpson Querrey Institute for Epigenetics, Northwestern University, Chicago, IL, USA. [4] Robert H. Lurie Comprehensive Cancer Center, Northwestern University, Chicago, IL, USA. ✉email: xiaomin.bao@northwestern.edu

An average person is estimated to turn over about 330 billion cells in a single day, from a spectrum of self-renewing somatic tissues such as blood, gut epithelium, and skin epidermis[1]. Maintenance of progenitor function is vital for supporting the continuous tissue regeneration, safeguarding tissue architecture and function. Several epigenetic regulators, including the DNA methyltransferase DNMT1 and the polycomb repressor EZH2, are recognized as key regulators supporting progenitor maintenance. DNMT1 and EZH2 are highly expressed in the progenitors to repress terminal differentiation, and they are downregulated in terminal differentiation[2–6]. Aberrant over-expression of DNMT1 and EZH2 is observed in cancer[7–9]. The upstream regulatory mechanisms sustaining the expression of DNMT1 and EZH2 in progenitors, however, remains poorly understood.

Nucleoporins (NUPs) are generally considered structural components of the nuclear-pore complexes (NPC), the essential gateways for nucleocytoplasmic transport. Emerging evidence supports that many NUPs are multi-functional proteins, participating in other cellular activities including chromatin binding and transcription regulation[10,11]. Several NUPs have been implicated in the regulation of stem cell function and differentiation, including NUP153, NUP50, and NUP98[12–14]. In the context of gene regulation, the NUPs can associate with many other transcriptional regulators. For example, NUP98 recruits the histone lysine methyltransferase Set1a to establish a subset of H3K4me3 marks in mouse hematopoietic progenitor maintenance[14]. In addition to Set1a, NUP98's interactions with p300, HDAC, RAE1, and CRM1 have been reported in other contexts[15]. Whether different NUPs can function synergically in the context of gene regulation remains incompletely understood. When and how the NUPs selectively cooperate with other transcriptional regulators to modulate specific groups of genes in a given biological context requires further investigation. As NUPs don't have canonical DNA- or histone-binding domains, how the NUPs can bind to chromatin at specific sites also remains largely unclear.

The human skin epidermis, a type of self-renewing epithelial tissue, is a highly accessible platform for investigating the gene regulatory mechanisms governing progenitor self-renewal and differentiation. The skin epidermal cells, known as keratinocytes, can be expanded ex vivo in the progenitor state, retaining their full potential to regenerate full-thickness epidermal tissue[16]. These progenitor-state keratinocytes can also be induced to the differentiation state using a combination of confluency and high calcium, recapitulating the gene expression kinetics in epidermal tissue differentiation[17,18]. The expandability of keratinocytes eases the incorporation of genomic approaches that sometimes require millions of cells in specific applications[19,20]. The regenerative capacity of these keratinocytes to form epidermis also enables the integration of genetics in three-dimensional human tissue[21,22]. However, the roles of NUPs in regulating the epidermal progenitor self-renewal and differentiation processes remain largely unclear.

In this study, we identified that NUP98 and RAE1 exist in a distinct complex in the soluble nuclear fraction of the progenitor-state keratinocytes. Both NUP98 and RAE1 are highly expressed in the progenitor state, and they are downregulated in the process of keratinocyte differentiation. Knockdown of NUP98 or RAE1 was sufficient to abolish the progenitor regenerative capacity. Mechanistically, we found that both NUP98 and RAE1 are essential for repressing differentiation and for sustaining progenitor self-renewal, including sustaining the expression of both DNMT1 and EZH2. We identified using ChIP-seq that NUP98 binds directly near the transcription start sites of these epigenetic regulators in the progenitor state. Furthermore, the NUP98 chromatin-binding sites are co-occupied by HDAC1. HDAC inhibition dysregulates the shared gene targets between NUP98 and RAE1 and abolishes NUP98 chromatin binding. Unexpectedly, we found that NUP98 and RAE1 recruit each other to the nucleolus upon HDAC inhibition. Taken together, our findings identified key regulatory roles contributed by the cooperation among NUP98, RAE1 and HDAC in progenitor maintenance, acting upstream to sustain the expression of key self-renewal regulators including both DNMT1 and EZH2. Our findings also highlight the roles of HDAC in facilitating NUP98 chromatin targeting in the progenitor state, antagonizing the nucleolar aggregation of both NUP98 and RAE1.

## Results

**NUP98 and RAE1 are enriched in progenitors and constitute a distinct complex.** Between the progenitor-state versus the differentiation state keratinocytes, we identified that 4 out of 5 nuclear-basket NUPs are significantly downregulated in differentiation (Fig. 1a), leveraging the RNA-seq data that we recently generated[20]. To determine how the enrichment of these nuclear-basket NUPs may influence progenitor maintenance, we asked if these NUPs exist in other complexes inside the nucleus. We extracted the soluble fractions from the nuclei of progenitor-state keratinocytes and performed size-exclusion chromatography. While TPR and NUP153 eluted in a range of earlier fractions, corresponding to larger protein-complex sizes up to 2MDa, NUP98 and RAE1 only eluted in the later fractions around the 160 KDa marker (Fig. 1b). The co-elution of NUP98 and RAE1, in the same fractions corresponding to small protein complexes, suggests that these two proteins may associate with each other independent of each other NUPs. To test this, we performed co-immunoprecipitation using nuclear extraction from progenitor-state keratinocytes. The NUP98 antibody co-immunoprecipitated both NUP98 and RAE1; the RAE1 antibody also co-immunoprecipitated both RAE1 and NUP98. Both co-immunoprecipitations did not enrich other nucleoporins such as TPR (Fig. 1c, d). Since RNA could mediate NUP98's interactions with other proteins[23], we investigated this by comparing NUP98-RAE1 co-immunoprecipitations with or without RNase treatment. Interestingly, the association between NUP98 and RAE1 was minimally affected by RNase (Supplementary Fig. 1a), suggesting that the NUP98-RAE1 interaction does not require an RNA component in this context. In addition, we probed the immunoprecipitated proteins by the NUP98 or RAE1 antibody using mAb414, which recognizes several FG-domain-containing NUPs. Although mAb414 detected multiple bands in the input lysate, this banding pattern was not observed in the immuno-precipitation by the NUP98 or RAE1 antibody (Supplementary Fig. 1b). These data suggest that NUP98 and RAE1 may play a role in the progenitor state independent of other NUPs.

To better understand the temporal expression of NUP98 and RAE1, transitioning from the progenitor state towards terminal differentiation, we performed both qRT-PCR and western blotting to quantify their mRNA and protein levels in the time course of calcium-induced keratinocyte differentiation. Significant reduction of NUP98 and RAE1 was detected on Day 2 (early differentiation), and this reduction extended to Day 4 (mid differentiation) of the differentiation time course (Fig. 1e–j). These data indicate that the downregulation of NUP98 and RAE1 is an early event in keratinocyte differentiation, which may play a role in regulating the switch from the progenitor state towards differentiation.

**NUP98 or RAE1 knockdown impairs progenitors' regenerative capacity.** To determine if the downregulation of NUP98 or RAE1 promotes the switch from the progenitor state toward

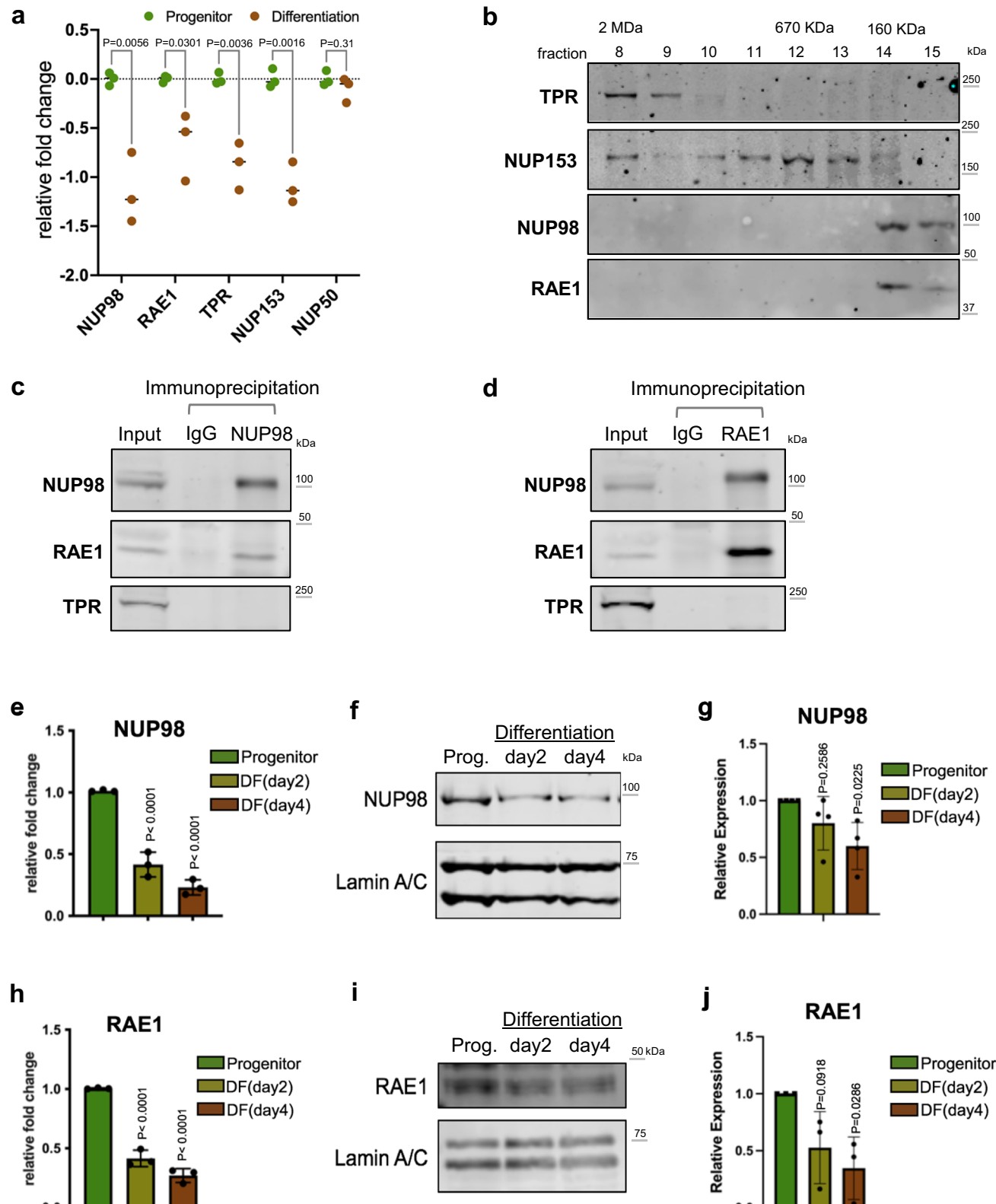

differentiation, we leveraged shRNA-mediated knockdown in the progenitor-state keratinocytes, tuning their expression down to a level comparable to the differentiation state. Three independent shRNAs for NUP98 or RAE1 were validated at the mRNA and protein levels (Fig. 2a, b, Supplementary Fig. 2a–d). Reduction of either NUP98 or RAE1 by these 6 shRNAs individually was sufficient to diminish keratinocyte clonogenicity (Fig. 2c–f). To evaluate the roles of NUP98 or RAE1 in influencing progenitors' regenerative capacity, we performed progenitor competition in skin epidermal regeneration. A 50:50 mix of GFP- or DsRed-expressing progenitor-state keratinocytes were seeded onto devitalized human dermis, raised in liquid-air interface. This organotypic regeneration process completes in a week, forming architecturally faithful human skin epidermis[18,19]. The

**Fig. 1 NUP98 and RAE1 are enriched in the progenitor state and constitute a distinct complex. a** Relative mRNA expression of nuclear-basket NUPs, comparing the progenitor-state versus the differentiated (day 4) primary human keratinocytes, based on RNA-seq data (multiple unpaired t test, N = 3 biological replicates). **b** Western blots showing the distribution of nuclear-basket NUPs in the fractions from size-exclusion chromatography (SEC), using the soluble extraction from the nuclei of progenitor-state keratinocytes. The fractions corresponding to the protein standards for SEC are labeled on the top. **c, d** Western blots showing the co-immunoprecipitation between NUP98 and RAE1 in the soluble extraction from the nuclei (progenitor-state keratinocytes). NUP98 and RAE1 co-immunoprecipitated each other, but not other nuclear-pore subunits such as TPR. **e** RT-qPCR comparing the relative NUP98 expression at the mRNA level in the progenitor-state, early- (day2) and mid- (day4) differentiation state of keratinocytes (one-way ANOVA with post-hoc test, N = 3 biological replicates, data are represented as mean ± standard deviation). **f, g** Western blots and quantifications comparing NUP98 protein expression in keratinocyte differentiation, with Lamin A/C used as the loading control (One-way ANOVA with post-hoc test, N = 4, data are represented as mean ± standard deviation). **h** RT-qPCR comparing the relative RAE1 mRNA expression in keratinocyte differentiation (one-way ANOVA with post-hoc test, data are represented as mean ± standard deviation). **i, j** Western blots and quantifications comparing RAE1 protein expression levels in keratinocyte differentiation, with Lamin A/C used as the loading control (one-way ANOVA with post-hoc test, N = 3, quantification data are represented as mean ± standard deviation).

GFP-expressing keratinocytes co-expressed control non-targeting shRNA; the DsRed-expressing keratinocytes co-expressed one of the three shRNAs: non-targeting control shRNA, NUP98-targeting shRNA, or RAE1-targeting shRNA. For the epidermis regenerated using keratinocytes both expressing the non-targeting control shRNAs, the red and green fluorescent keratinocytes showed comparable representation in the tissue, indicating that the expression of GFP or DsRed did not differentially influence the progenitors' regenerative capacity. In contrast, the red keratinocytes co-expressing NUP98 or RAE1 shRNA were outcompeted by the green keratinocytes co-expressing the control shRNA, with diminished representation in the basal progenitor compartment (Fig. 2g, h). These findings suggest that the high expression level of both NUP98 and RAE1 is essential for progenitor maintenance.

To identify the key cellular processes influenced by NUP98 or RAE1 knockdown, we performed transcriptome profiling using RNA-seq. In total, we identified 1493 significantly changed genes with NUP98 knockdown, and 1401 significant changed genes with RAE1 knockdown (p < 0.05, two tailed, Wald test, average fold change for three independent shRNAs >2, and individual shRNA fold change >1.5, Supplementary Data 1, Supplementary Data 2). These two sets share 597 genes (p = 1 × 10$^{-321}$, Fishers' exact test), which are altered in the same direction (upregulated or downregulated) with either NUP98 or RAE1 knockdown (Fig. 2i, j). Top gene ontology (GO) terms of the shared upregulated genes include epidermal development and keratinocyte differentiation; Top GO terms of the shared downregulated genes are related to cell division (Fig. 2k). We further identified that 77% of these shared genes are also significantly altered in calcium-induced differentiation (Supplementary Fig. S2e). In addition, we investigated if NUP98 or RAE1 knockdown was sufficient to trigger apoptosis, leveraging two different dyes that are sensitive for mitochondria potential. While the staining was abolished in the positive-control keratinocytes treated with $H_2O_2$, keratinocytes with NUP98 or RAE1 knockdown retained the staining similar to the control knockdown (Supplementary Fig. 2f, g), suggesting that the knockdown strategy did not trigger apoptosis. Thus, these findings suggest that the enrichment of NUP98 and RAE1 in the progenitor state is essential for sustaining proliferation and repressing differentiation.

**NUP98 binds near the TSSs of key regulators in progenitor maintenance.** NUP98 has been reported to bind to different genomic regions in the context of different cell types[14,24]. To investigate how NUP98 genomic binding could influence gene expression in epidermal progenitor maintenance, we performed NUP98 ChIP-seq in keratinocytes. In the progenitor-state keratinocytes, we identified a total of 1554 NUP98 ChIP peaks. In the differentiation state, however, NUP98 binding is reduced across all these regions. The majority (86%, 1334 peaks) of these NUP98 ChIP-seq peaks are unique for the progenitor state, but not for the differentiation state. Only 14% of these peaks (220 peaks) were also called in the differentiation state, yet the ChIP enrichment at these regions is also reduced in differentiation. A small number (117) NUP98 ChIP peaks were identified as unique to the differentiation state, most of which (74.3%) are located at least 10 kb away from the TSSs. In contrast, most peaks (71%) identified in the progenitor state are located within 3 kb from the transcription start sites (Fig. 3a–g, Supplementary Data 3). Thus, the switch from the progenitor state towards differentiation involved an overall reduction of NUP98 genomic binding especially near the transcription start sites (Fig. 3h).

We subsequently annotated these genes, which are associated with NUP98 ChIP-seq binding peaks in the progenitor state. Top GO terms are related to transcription co-regulator binding, chromatin binding, and transcription factor binding (Fig. 3i), suggesting that these genes could be upstream regulators of key biological processes. The intersection of the ChIP-seq targets with the NUP98 RNA-seq data identified a total of 101 Direct Target Genes. The majority of these 101 Direct Target Genes also show similar upregulation or downregulation with RAE1 knockdown (Fig. 3j). Interestingly, the downregulated Direct Target Genes include key regulators governing epidermal progenitor maintenance, including the DNA methyltransferase DNMT1 and its recruiter URHF1[6], the polycomb group protein EZH2[3] (Fig. 3k–o), in addition to the DNA replication regulators CDT1 and RRM2. These Direct Target Genes also include upregulated genes such as DUSP10 and JARID2, which are less well characterized in the context of keratinocyte differentiation (Supplementary Data 4).

To determine if RAE1 binds to chromatin together with NUP98, we generated a HA-tagged RAE1 construct and expressed it in keratinocytes, as the commercially available antibodies we screened did not yield high-quality ChIP-seq data. HA-RAE1 co-immunoprecipitated NUP98 in the progenitor-state keratinocytes (Supplementary Fig. 3a), confirming that the HA tag does not interfere with RAE1's association with NUP98. Using double-crosslinking ChIP-seq, we identified that RAE1 is enriched in 83% of the NUP98 ChIP-seq peak regions. The NUP98-RAE1 overlapping ChIP-seq peaks, but not the NUP98 unique peaks, are predominantly located near the TSSs (Supplementary Fig. 3b–f). Among the 101 direct target genes of NUP98, 96 of these genes are also associated with RAE1 ChIP-seq enrichment, including DNMT1 and EZH2 (Supplementary Fig. 3g–i). These findings suggest that NUP98 and RAE1 bind directly near the TSSs to regulate gene expression.

DNMT1 has been previously characterized as a key regulator for human epidermal progenitor maintenance[6]. We confirmed the drastic downregulation of DNMT1 in keratinocyte differentiation using western blotting (Supplementary Fig. 4a, b).

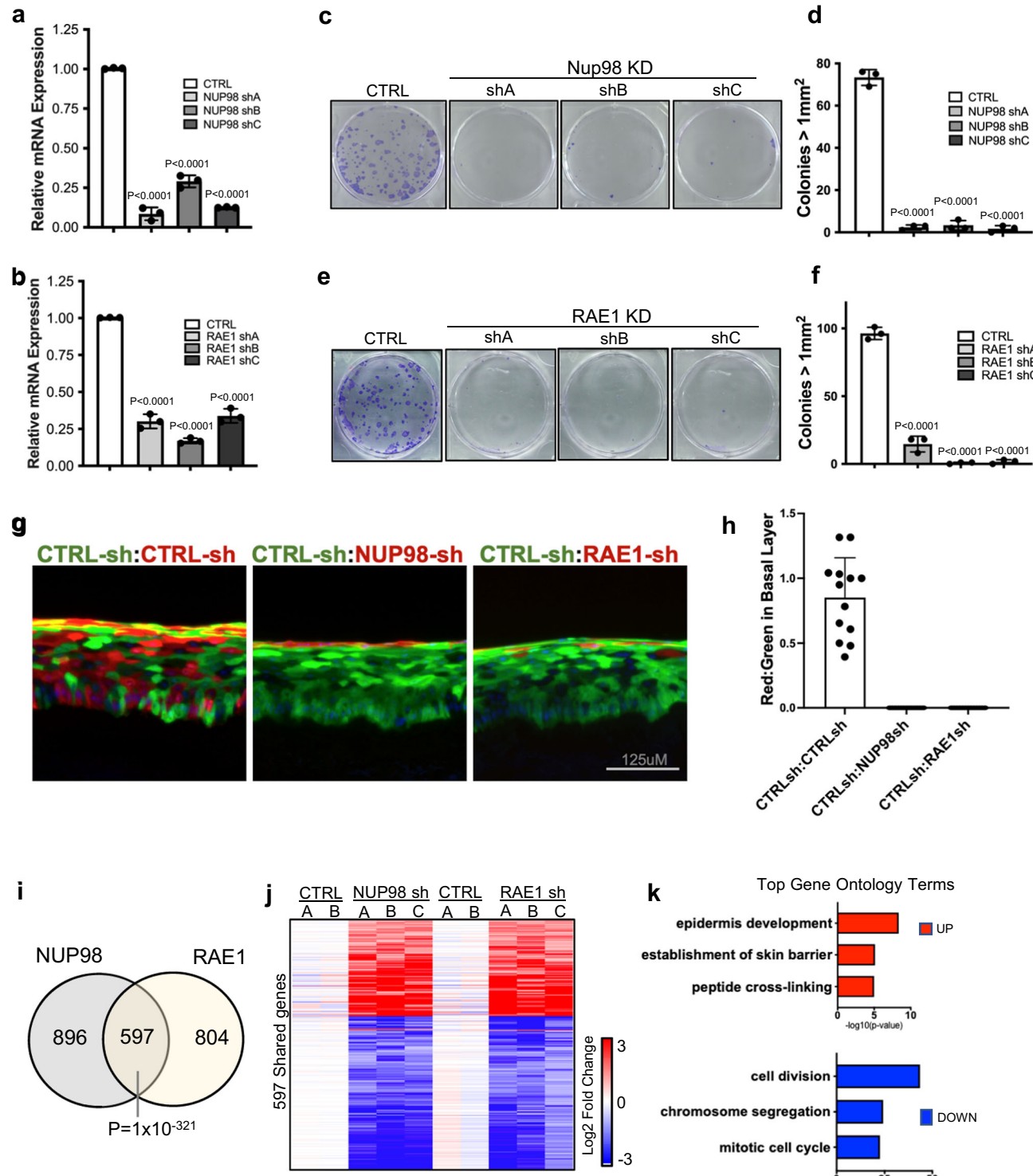

**Fig. 2 NUP98 or RAE1 knockdown impairs progenitors' regenerative capacity. a, b** RT-qPCR showing knockdown efficiency of 3 independent shRNA targeting NUP98 or RAE1 (one-way ANOVA with post-hoc test, N = 3, data are represented as mean ± standard deviation). **c–f** Representative images and quantification of clonogenicity assay comparing keratinocytes with NUP98 or RAE1 knockdown versus control (one-way ANOVA with post-hoc test, N = 3, quantification data are represented as mean ± standard deviation.) **g** Representative images of competition assay in epidermal tissue regeneration. An equal number of keratinocytes expressing DsRed or GFP were mixed seeded onto human dermis. GFP-expressing keratinocytes co-express control shRNA, and DsRed-expressing keratinocytes co-express control shRNA or shRNA targeting NUP98 or RAE1 (scale bar = 125 μm). **h** Quantification of DsRed or GFP-expressing keratinocytes in the basal layer of the regenerated epidermis, data are represented as mean ± standard deviation. **i** Venn diagram showing the overlap of NUP98 and RAE1 knockdown differentially expressed genes (Fisher's exact test, p = 1 × 10⁻³²¹). **j** Heatmap showing the relative expression of shared genes with NUP98 or RAE1 knockdown. **k** Top three gene ontology (GO) terms for the upregulated or downregulated genes shared by NUP98 and RAE1 knockdown, identified by RNA-seq analyses.

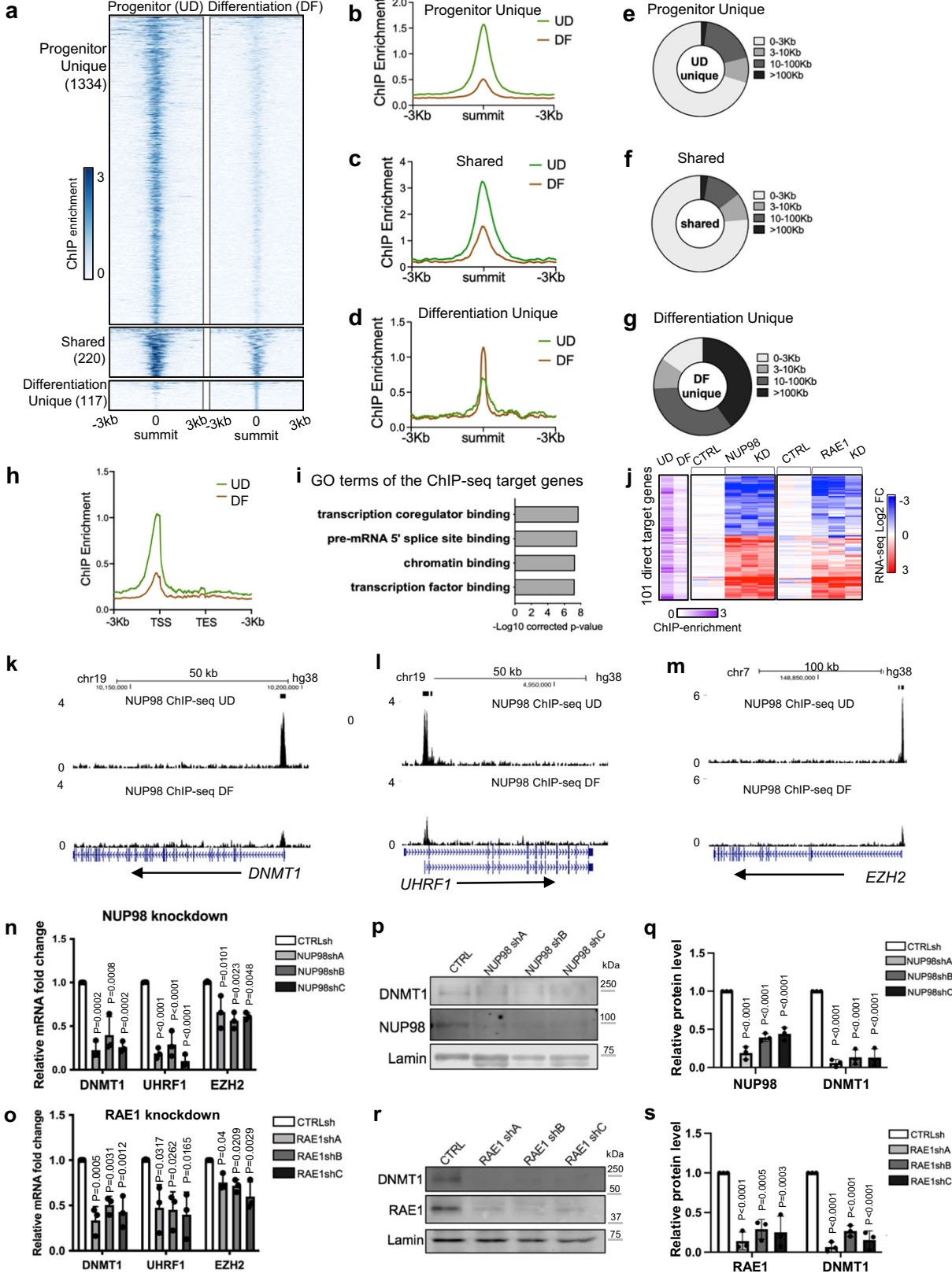

Leveraging the published transcriptome-profiling data of DNMT1 knockdown in progenitor-state keratinocytes[6], we compared the differentially expressed genes upon DNMT1 knockdown with the differentially expressed genes upon NUP98 or RAE1 knockdown. The intersection of these data sets identified a total of 236 shared genes (Supplementary Fig. 4c, d). The top GO terms of the upregulated shared genes are associated with epidermal development and keratinocyte differentiation; the downregulated genes are related to cell division and DNA replication (Supplementary Fig. 4e, f). We further validated using western blotting that the DNMT1 protein levels are drastically reduced with NUP98 or RAE1 knockdown, in keratinocytes cultured in the progenitor

**Fig. 3 NUP98 binds near the TSSs of key regulators in progenitor maintenance. a–d** Summit-centered heatmaps and average diagrams comparing NUP98 ChIP enrichment between the progenitor state (UD) and the differentiation state (DF). **e–g** Pie charts showing the distribution of the NUP98 ChIP-seq peaks, based on their distances to the nearest Transcriptional Start Sites (TSSs). **h** Average diagram comparing NUP98 ChIP-seq enrichment near the transcription start and end sites of the target genes. **i** Top Gene Ontology (GO) terms of the nearest genes associated with NUP98 ChIP-seq peaks. **j** Heatmap showing the 101 genes, featuring NUP98 ChIP-seq enrichment and are significantly changed with NUP98 knockdown. Relative NUP98 ChIP enrichment in UD and DF associated with these gene, and their relative expression with NUP98 or RAE1 knockdown, are included for each of these genes side by side. **k–m** Browser track examples of NUP98 ChIP-seq enrichment, comparing UD vs. DF. **n, o** qRT-PCR validation of representative target genes with NUP98 or RAE1 knockdown (one-way ANOVA with post-hoc test, N = 3 biological replicates, data are represented as mean ± standard deviation). **p–s** Western blot and quantification comparing DNMT1 protein levels with NUP98 or RAE1 knockdown, with Lamin used as the loading control (one-way ANOVA with post-hoc test, N = 3, quantification data are represented as mean ± standard deviation).

condition (Fig. 3p–s). Thus, DNMT1 downregulation at least partially accounts for the differentiation induction and proliferation inhibition observed with NUP98 or RAE1 knockdown. Taken together, these data suggest that the chromatin binding of NUP98 and RAE1 is involved in progenitor maintenance, through directly controlling the expression of key proliferation/differentiation regulators such as DNMT1.

**NUP98 co-localizes with HDAC1 on chromatin and cooperates with HDAC in gene regulation.** Given NUP98's chromatin binding to genes encoding key regulators of the proliferation/differentiation process, we investigated potential mechanisms facilitating NUP98's chromatin binding to these specific genomic regions. Motif search did not uncover specific transcription factors that can explain the majority of the genomic binding sites. We then compared NUP98 ChIP-seq peaks with the ChIP-seq peak files of other transcriptional regulators and histone marks generated using the same cell type of primary human keratinocytes (Fig. 4a, Supplementary Data 5). Consistent with NUP98's binding near the transcription start sites, NUP98 ChIP-seq peak regions are heavily (>80%) co-occupied by the histone marks (H3K4me2/3, H3K27Ac, H3K9Ac) and Pol II. Interestingly, HDAC1 also co-occupies 82% of all NUP98 ChIP-seq peaks, comparable to Pol II (85%) and much higher than the lineage-specific transcription factor p63 (29%). Similar to NUP98, HDAC1 also binds near the TSSs of the NUP98 target genes, such as DNMT1 and EZH2 (Fig. 4b–d). Thus, HDAC1 stood out as a candidate that could cooperate with NUP98 in chromatin binding and gene regulation.

This extensive overlap of NUP98 and HDAC1 ChIP-seq peaks suggested that these two proteins could physically associate with each other. We confirmed this using crosslinking immunoprecipitation, that the NUP98 or RAE1 antibody co-immunoprecipitated HDAC1 in keratinocyte lysate (Fig. 4e, f). To determine if HDAC influences gene expression similar to NUP98 and RAE1, we leveraged the HDAC inhibitors Romidepsin (ROM) and SAHA. When added to keratinocytes cultured in the progenitor state, these inhibitors consistently downregulated representative NUP98 direct target genes, such as DNMT1, UHRF1 and EZH2 (Fig. 4g). Using RNA-seq, we further identified that 74% of NUP98-RAE1 target genes are also significantly altered by HDAC inhibition (Fig. 4h, i, Supplementary Data 6). Since p300 was also identified as an interacting protein with NUP98-fusion proteins in hematopoietic malignancies[25], we found that p300 inhibition did not drastically alter NUP98's target gene expression in keratinocytes (Supplementary Fig. 5a), supporting that that HDAC is specifically involved in modulating the target genes controlled by NUP98 and RAE1 in epidermal progenitor maintenance.

Building on the gene expression changes of NUP98 and RAE1's target genes induced by HDAC inhibition, we further investigated the roles of HDAC1. We designed and validated a total of 3 independent shRNAs targeting HDAC1. Interestingly, all three of these shRNAs consistently downregulated the target genes of NUP98 and RAE1, such as DNMT1 and UHRF1 (Supplementary Fig. 5b). Thus, HDAC1 is involved in regulating the NUP98 and RAE1's target gene expression in the progenitor state. In the keratinocyte differentiation process, HDAC1's protein level is slightly reduced, with an average of 54% relative expression on differentiation day 4 as compared to the progenitor state (Supplementary Fig. 5c). Consistently, the HDAC1 ChIP enrichment was only moderately reduced in the differentiation state (Supplementary Fig. 5d). We further compared HDAC1 ChIP-seq enrichment in NUP98 binding sites between the progenitor state versus the differentiation state (day 4, Supplementary Fig. 5e–h). The majority (92.1%) of these sites did not show drastic reduction of HDAC1 binding in differentiation. Only in a very small fraction (7.7%) of these sites, HDAC1 enrichment showed significant reduction (fold change >2, p < 0.05). Thus, HDAC1 ChIP enrichment was only modestly reduced in differentiation in the NUP98 binding sites, in contrast to the drastic reduction of NUP98 in these sites in differentiation, suggesting that NUP98 is not required for maintaining HDAC1's chromatin binding in their shared binding sites.

**NUP98 chromatin binding is dependent on HDAC activity.** Given the overlap between NUP98 and HDAC1 in genomic binding sites and in gene regulation, we asked if HDAC activity functions upstream to influence NUP98 genomic binding. We performed NUP98 ChIP-seq in keratinocytes with or without HDAC inhibition. Remarkably, the NUP98 ChIP-seq signals were diminished with HDAC inhibition using either SAHA or Romidepsin (Fig. 5a–c), suggesting that NUP98's genomic binding to its target genes is dependent on HDAC activity.

To determine if NUP98's subcellular localization was also affected by HDAC inhibition, we performed NUP98 immunofluorescence staining in keratinocytes with HDAC inhibition as compared to DMSO control. While NUP98 showed diffused staining inside the nucleus in the control condition, in addition to its nuclear periphery enrichment, HDAC inhibition by SAHA or Romidespin consistently resulted in 1–2 aggregated foci of NUP98 in each nucleus. Co-staining with the nucleolus marker fibrillin confirmed that the NUP98 localized to the nucleolus (Fig. 5d, e). Similar to NUP98, RAE1 also aggregated to the nucleolus upon HDAC inhibition (Fig. 5f, g).

A recent paper described FUS targeting to the nucleolus under transcriptional stress[26]. We asked if FUS is involved in NUP98 and RAE1's nucleolus enrichment. Co-Immunofluorescence staining was performed for both NUP98 and FUS with HDAC inhibition, as compared to the control. While NUP98 was consistently enriched in the nucleolus with HDAC inhibition, no FUS enrichment was detected (Supplementary Fig. 6). These data suggest that NUP98 and RAE1's nucleolar targeting is independent of the mechanisms involved in targeting FUS to the nucleolus.

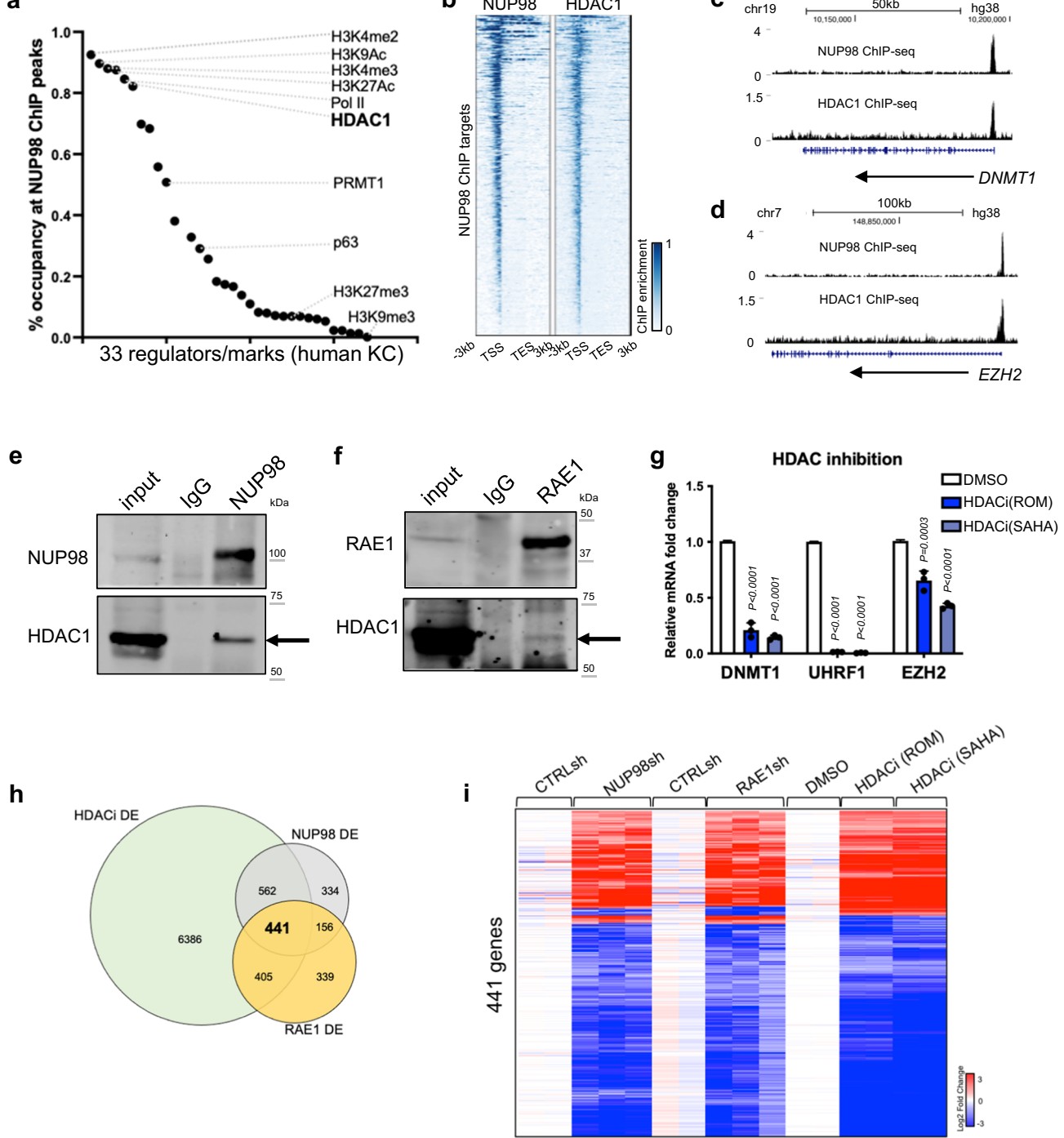

**Fig. 4 NUP98 co-localizes with HDAC1 on chromatin and cooperates with HDAC in gene regulation. a** Occupancy of other epigenetic marks or regulators in NUP98 ChIP-seq peak regions. **b** Heatmap comparing NUP98 ChIP-seq and HDAC1 ChIP-seq on NUP98 peaks. **c, d** Browser tracks showing the co-occupancy of NUP98 and HDAC1 on representative target genes. **e, f** Crosslinking co-immunoprecipitation using NUP98 or RAE1 antibody to detect HDAC1. **g** qRT-PCR showing the relative expression of RAE1-NUP98 target genes with HDAC inhibition (one-way ANOVA with post-hoc test, N = 3, data are represented as mean ± standard deviation). **h** Venn diagram showing the overlap of NUP98, RAE1, and HDACi differentially expressed genes. **i** Heatmap showing the expression of the differentially expressed genes shared among NUP98 knockdown, RAE1 knockdown, and HDAC inhibition.

**NUP98 and RAE1's nucleolar localization is balanced by HDAC and HAT activities**. The nucleolar enrichment of NUP98 and RAE1, induced by HDAC inhibition, suggests that protein acetylation could be involved in this process. To test this, we designed an experiment to inhibit protein acetylation in combination with HDAC inhibition, leveraging the p300/CBP HAT inhibitors A485 and C646[27,28]. As shown in Fig. 6 and

Supplementary Fig. 7, HAT inhibition alone did not significantly alter the subnuclear localization of NUP98 or RAE1; however, when the keratinocytes were pre-treated with HAT inhibitors for 6 h, HDAC inhibition was no longer able to induce NUP98 or RAE1's nucleolar localization. This antagonism between HAT inhibition and HDAC inhibition was consistently observed among the two independent p300/CBP inhibitors (A485 or

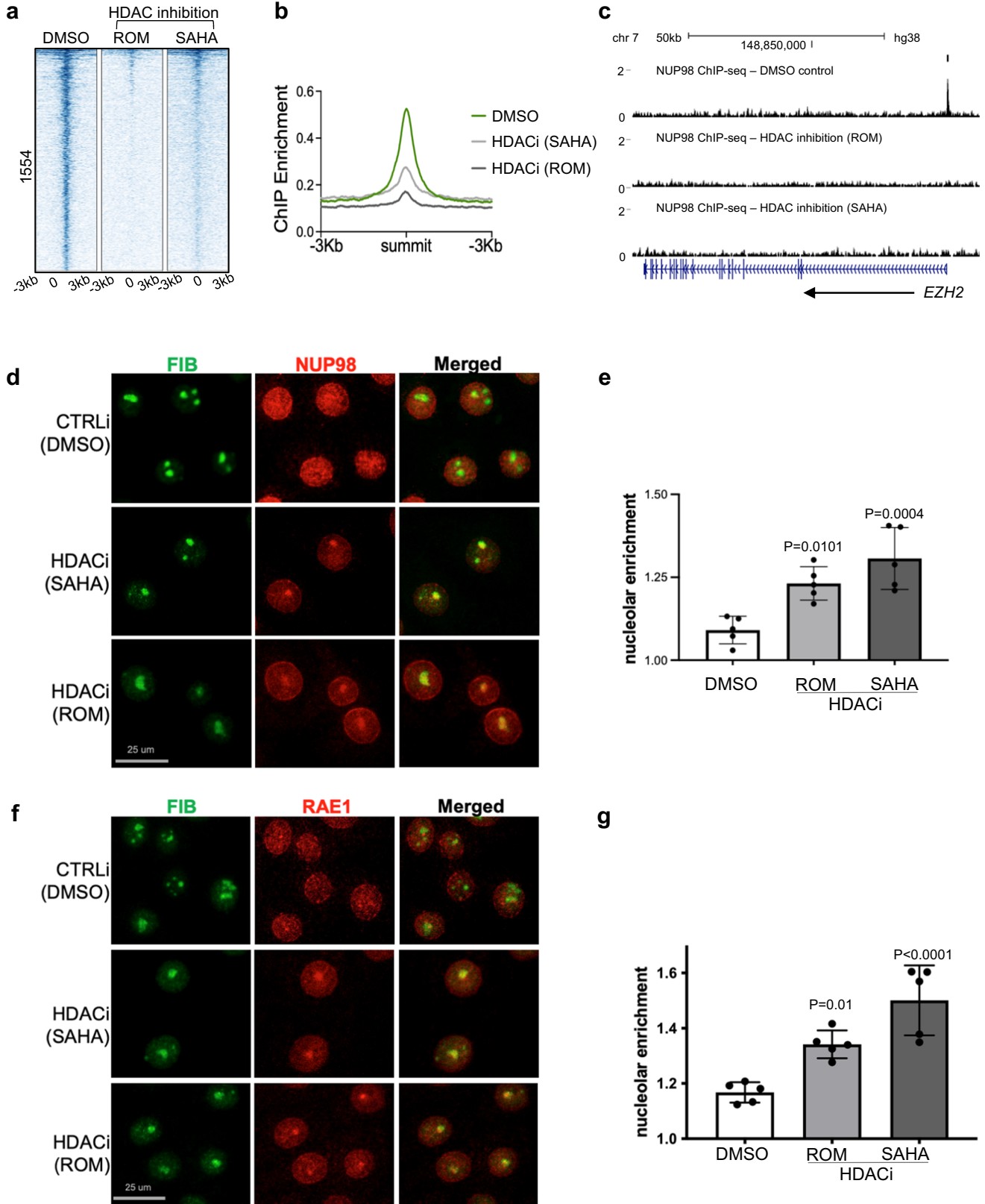

C646). These data support a role of protein acetylation in mediating the nucleolar localization of NUP98 or RAE1, as a consequence of HDAC inhibition.

**Interdependence between NUP98 and RAE1 in nucleolar localization upon HDAC inhibition.** Given the association between NUP98 and RAE1, we further asked if NUP98's nucleolar targeting upon HDAC inhibition depends on RAE1. While keratinocytes expressing control non-targeting shRNA still showed NUP98 aggregation with HDAC inhibition, this nucleolar aggregation was abolished in keratinocytes expressing shRNAs targeting RAE1 (Fig. 7a, b). Thus, the nucleolar

**Fig. 5 NUP98 chromatin binding is dependent on HDAC activity. a** Heatmap showing NUP98 ChIP enrichment with HDAC inhibition (using ROM or SAHA) as compared to DMSO control. **b** Average profile showing NUP98 ChIP enrichment with HDAC inhibition (using ROM or SAHA) as compared to DMSO control. **c** Browser track showing a representative target gene *EZH2*, comparing NUP98 ChIP enrichment with HDAC inhibition versus the DMSO control, in keratinocytes cultured in the progenitor state. **d** Representative images showing keratinocytes co-immunostained with antibodies targeting NUP98 and the nucleolus marker Fibrillarin (FIB), comparing HDAC inhibition versus DMSO control (scale bar = 25 μm). **e** Quantification of the relative NUP98 enrichment in the nucleolus in HDAC inhibition versus the DMSO control (one-way ANOVA with post-hoc test, N = 5, data are represented as mean ± standard deviation). **f**, **g** Co-immunostaining of RAE1 and FIB in keratinocytes treated with HDAC inhibitors versus the DMSO control (scale bar = 25 μm), and quantification (one-way ANOVA with post-hoc test, N = 5, quantification data are represented as mean ± standard deviation).

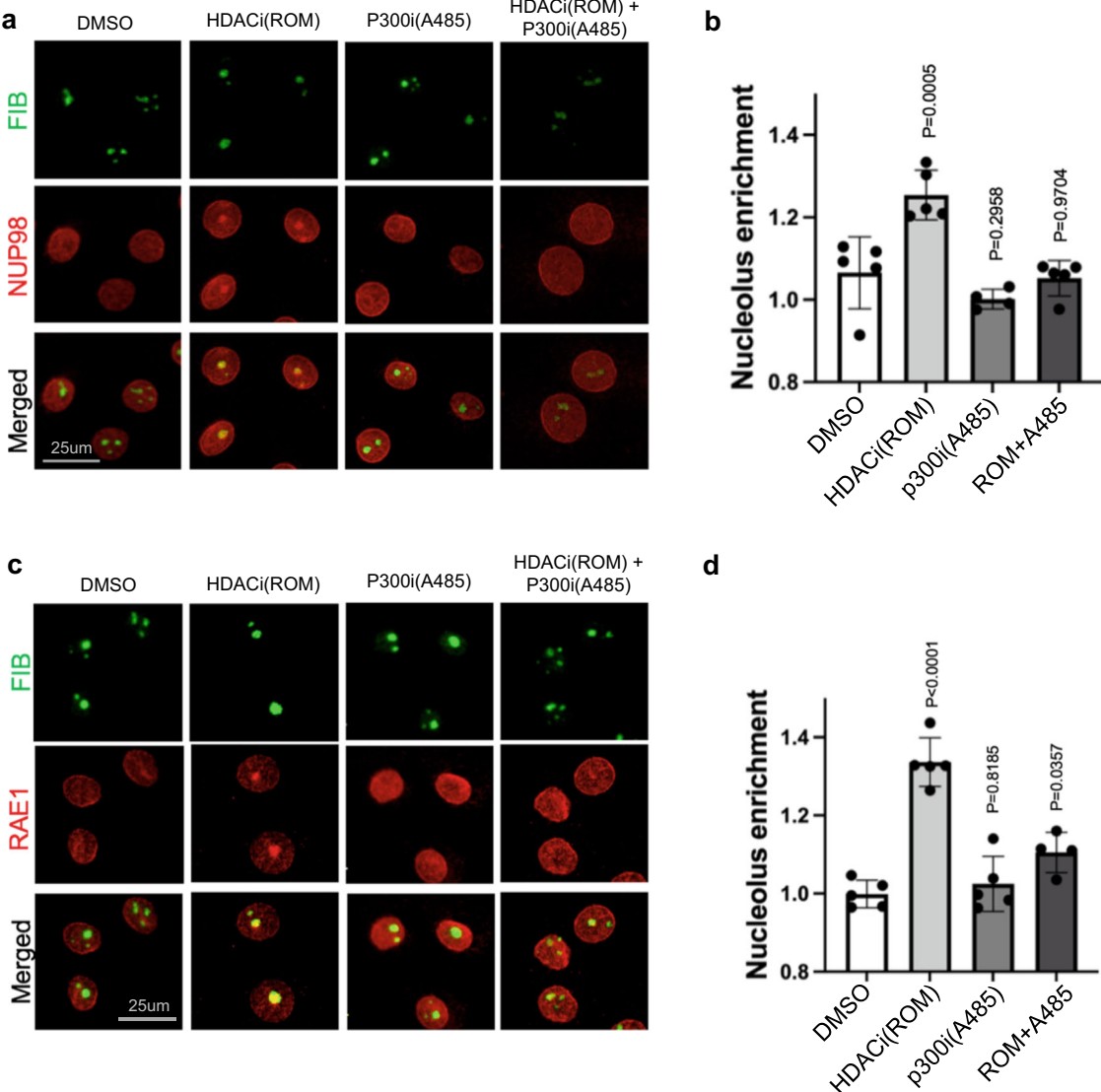

**Fig. 6 NUP98 and RAE1's nucleolar localization is balanced between HDAC and HAT activities. a** Co-immunostaining of NUP98 and nucleolus marker FIB in keratinocytes treated with HDAC inhibitor, ROM, and/or the HAT inhibitor A485 (scale bar = 25 μm). **b** quantification of NUP98 nucleolus enrichment in these treatment conditions (one-way ANOVA with post-hoc test, N = 5, data are represented as mean ± standard deviation). **c** Co-immunostaining of RAE1 and the nucleolus marker FIB in keratinocytes treated with HDAC inhibitor, ROM, and/or the HAT inhibitor A485 (scale bar = 25 μm). **d** quantification of RAE1 nucleolus enrichment among these conditions (one-way ANOVA with post-hoc test, N = 5, data are represented as mean ± standard deviation).

enrichment of NUP98 requires the intact function of RAE1. We subsequently investigated if RAE1's nucleolar targeting upon HDAC inhibition also depended on NUP98. We treated keratinocytes expressing either NUP98 shA, NUP98 shB or control non-targeting shRNA with the HDAC inhibitor ROM. RAE1's nucleolar targeting was abolished in keratinocytes expressing NUP98 shRNA, but not the non-targeting control. Using two independent shRNAs targeting NUP98, RAE1 also did not target to the nucleolus (Fig. 7c, d). RAE1 knockdown alone did not induce NUP98 nucleolar localization, and NUP98 knockdown did not induce RAE1 nucleolar localization, without HDAC inhibition (Supplementary Fig. 8). These findings indicate that NUP98 and RAE1 depend on each other to localize to the nucleolus upon HDAC inhibition.

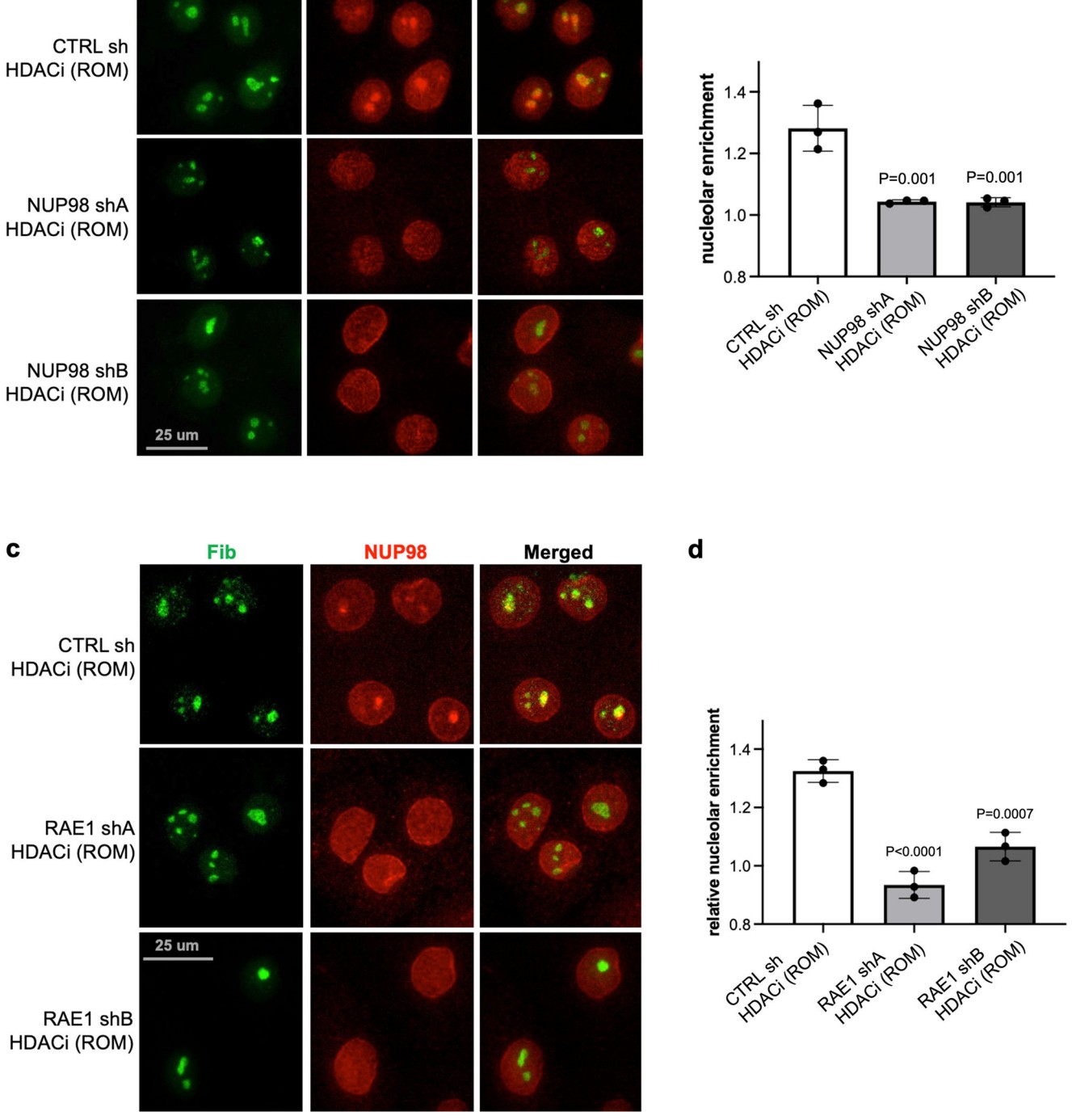

**Fig. 7 Interdependence between NUP98 and RAE1 in nucleolar localization upon HDAC inhibition. a**, **b** Representative images showing co-immunostaining of RAE1 and the nucleolus marker FIB in keratinocytes treated with NUP98 knockdown versus control knockdown (scale bar = 25 μm), and quantifications of RAE1 nucleolar enrichment among these conditions (one-way ANOVA with post-hoc test, N = 3, quantification data are represented as mean ± standard deviation). **c**, **d** Co-immunostaining of NUP98 and the nucleolus marker FIB in keratinocytes treated with RAE1 knockdown versus control knockdown (scale bar = 25 μm), and quantifications of NUP98 nucleolar enrichment among these conditions (one-way ANOVA with post-hoc test, N = 3, quantification data are represented as mean ± standard deviation).

Taken together, these findings suggest a model that NUP98 and RAE1 promote progenitor maintenance by directly binding to and controlling the expression of key epigenetic regulators under two conditions: elevated expression and HDAC activity. The elevated expression in the progenitor state allows the presence of a soluble intranuclear pool of NUP98 and RAE1, allowing chromatin binding in addition to their nuclear-pore association; HDAC activity is also essential for antagonizing the nucleolar targeting for NUP98 and RAE1, promoting NUP98 genomic targeting to the key epigenetic regulators for self-renewal (Fig. 8).

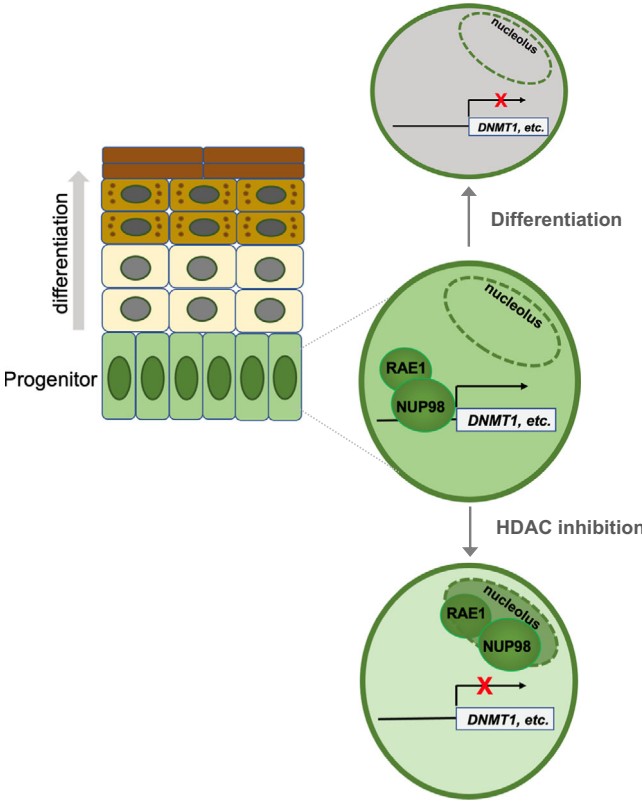

**Fig. 8 Working model.** NUP98 and RAE1 support epidermal progenitor maintenance through binding to chromatin near the transcription start sites of key target genes such as the epigenetic repressor DNMT1. NUP98 and RAE1's roles in progenitor maintenance depend on two factors: high expression and HDAC activity. The high expression of NUP98 and RAE1 provides the chromatin-binding pool, in addition to their nuclear-pore incorporation; HDAC activity supports NUP98's binding on chromatin near the TSSs of key target genes, antagonizing the nucleolar localization of NUP98 and RAE1.

## Discussion

Our findings highlight the regulatory roles of NUP98 and RAE1 in epithelial progenitor maintenance. NUP98 and RAE1 are expressed at higher levels in the progenitor state than the differentiation state, and their high expression in progenitors is essential for sustaining proliferation and suppressing differentiation. Mechanistically, we find that NUP98 and RAE1 exist in a separate complex in the soluble fraction of the nucleus. In particular, NUP98 directly binds near the transcription start sites of key regulators governing progenitor self-renewal, including DNMT1, EZH2, and UHRF1. We further uncovered that NUP98's chromatin association is dependent on HDAC activity. HDAC inhibition diminished NUP98's chromatin binding, and resulted in nucleolar accumulation of NUP98 and RAE1. Thus, these data demonstrated multiple levels of crosstalk between the epigenetic regulators and the nucleoporins in progenitor maintenance.

Similar to several other transcriptional regulators, such as CPSF and PRMT1[20,22], NUP98 and RAE1 are expressed at a higher level in the progenitor state as compared to the differentiation state. In the case of CPSF, its higher expression allows it to associate with RNA-binding proteins and participates in gene regulation, in addition to its housekeeping functions in processing RNA after their full-length synthesis[20]. Similarly, we speculate that the high expression of NUP98 and RAE1 are necessary for supplying a chromatin-binding pool, in addition to their essential roles of constituting the nuclear-pore complex.

Our data placed NUP98, RAE1 and HDAC upstream to influence the expression from several key regulators of progenitor maintenance, such as DNMT1, UHRF1 and EZH2. Several transcription factors have been implicated in promoting the expression of DNMT1, including SP1, STAT3, E2F, and MTF-1[29–32]. Interestingly, the expression of DNMT1 and UHRF1 were found to be sensitive to the MEK inhibitors in cancer cells[33], suggesting that the EGRF pathway plays a role in sustaining the expression of these two regulators. Using a combination of ChIP-seq and RNA-seq, we found that NUP98 and HDAC1 are essential for sustaining the expression of DNMT1 and URHF1 in epidermal progenitors. This was further confirmed using shRNAs targeting HDAC1. Considering that NUP98 chromatin binding profiles can differ drastically among different cell types[14,24], it would be interesting to determine in the future if specific transcription factors or pathways cooperate with both NUP98 and HDAC in transcriptional regulation in the context of epidermal progenitors. Although HDAC1 can associate with the epidermal lineage-specific transcription factor p63 in regulating embryonic skin development in mouse[34], we found minimal overlap between p63 and NUP98 in ChIP-seq peaks. At this moment, we cannot exclude a possibility that p63 could still be involved in participating the gene regulation controlled by NUP98 and RAE1, through potential promoter-enhancer interactions.

A striking observation from this study is the nucleolar localization of both NUP98 and RAE1 upon HDAC inhibition, in keratinocytes cultured in the progenitor state condition. Since HAT inhibition abolished the nucleolar localization induced by HDAC inhibition, these findings suggest that protein acetylation is involved in regulation this nucleolar localization. The key substrates of acetylation in this process remain unclear, and it is entirely possible that multiple substrates could be involved in influencing NUP98 and RAE1's subnuclear localization. The nucleolar localization of NUP98 was previously observed in a different context with actinomycin D or a-amanitin treatment[35,36]. In addition to NUP98, actinomycin D treatment or Pol I knockdown was sufficient to localize the NUP98-interacting-protein CRM1 to the nucleolus. This nucleolar enrichment of NUP98 and CRM1 by actinomycin D was abolished with the addition of the CRM1 inhibitor LMB, or by knocking down the ribosome-export-receptor NMD3[37]. Furthermore, ribosomal transcription is regulated by protein acetylation, involving the competitive actions of HDAC and HAT[38]. Taken together, these data suggesting that NUP98 and RAE1's nucleolar localization, induced by HDAC inhibition, is likely to be involved in the ribosomal biogenesis and export pathway.

Our findings also shed new light on the mechanisms underlying how NUP98 associates with chromatin. NUP98 does not possess any DNA or histone-binding domains, yet it can associate with hundreds of specific genomic regions in several cell types, and multiple NUP98-interacting proteins have been identified[23]. Although NUP98-fusion proteins in leukemia have been linked to HDAC binding[39,40], in addition to many other transcription regulators including p300, it remained incompletely understood regarding how HDAC influences NUP98 function. In the progenitor-state keratinocytes, we found that HDAC inhibition, but not p300 inhibition, phenocopied the NUP98 or RAE1 in modulating self-renewal and differentiation gene expression. This agrees with the previous findings in leukemia that HDAC inhibition de-represses the target genes of NUP98-fusion protein[41], and RAE1 contributes to the leukemogenesis driven by NUP98 fusions[42]. Our ChIP-seq data analyses identified that HDAC1 co-occupies NUP98 chromatin binding sites. We further uncovered that HDAC inhibition drastically reduced NUP98 chromatin

binding. These data demonstrate that NUP98 binds to chromatin in an HDAC-dependent manner. HDAC inhibitors was suggested as a therapeutic strategy for NUP98-fusion AML patients[39]; however, the NUP98-fusion events are restricted in the blood lineage for these patients, and normal NUP98 function is expected in other somatic tissues. Our findings demonstrated that HDAC inhibition directly impacts wild-type NUP98's roles in epithelial progenitor maintenance. Future work improving our understanding of the action of NUP98 in normal tissue home-ostasis, and its differences from the NUP98 oncogenic fusions, will inform better therapeutic design with minimized side effects. Since other NUPs such as NUP62 has been shown to modulate squamous cell carcinoma proliferation and differentiation[43], it would also be interesting to investigate how different NUPs cooperate to modulate epithelial homeostasis and pathogenesis.

## Methods

This study using primary human keratinocytes was reviewed by Northwestern University IRB (Institutional Review Board) and was determined as not human research. Surgically discarded neonatal foreskin was obtained from Northwestern SBDRC (Skin Biology & Diseases Resource Based Center), and tissue collection was approved by IRB (#STU00009443) with all relevant ethical regulation followed and informed consent obtained.

**Primary keratinocyte culture**. Primary keratinocytes isolated from 6 to 7 different de-identified donors were pooled for all the experiments associated with this study. To culture and maintain the keratinocytes in the undifferentiated condition, a 50:50 mix of two culture media, KSFM (Gibco, #17005-142) and Medium 154 (Gibco #M-154-500), was used. For calcium-induced differentiation, the kerati-nocytes were seeded in confluency with the addition of 1.2 mM $CaCl_2$ for the duration of 4 days.

**Lentiviral- or retroviral-mediated gene delivery**. For lentivirus production, HEK293T cells were transfected with 4 μg of shRNA or protein-expression plasmid in 6-cm plates using the lipofectamine 3000 kit (Thermo Fisher) together with pCMV-dR8.91 (3 μg) and pUC-MDG (1 μg) helper plasmids. For retrovirus pro-duction, Phoenix cells were transfected with 8 μg of DNA in 6-cm plates, for retrovirus production. Viral supernatant was used to infect keratinocytes at an appropriate viral dilution, with spin infection at 1250 rpm for 1 h at 32 °C with 5 mg/mL polybrene. Two days after spin infection, keratinocytes were selected with 2 μg/mL puromycin for 48 h.

**Plasmid construction**. The RAE1 gene was cloned with a N-terminal HA tagged into pLZRS. Gene transfer was performed by retroviral delivery. For NUP98, RAE1, and HDAC1 knockdown, shRNAs were designed using BLOCK-it RNAi designer (Invitrogen) and cloned into pLKO. Two control shRNA constructs, pLKO-NT (non-targeting) and pLKO-GFP, were generous gifts from Dr. Ali Shilatifard's laboratory. Oligo sequences are included in Supplementary Data 7.

**Inhibitor treatment**. HDAC inhibitors romidepsin (ROM, ApexBIO) and Vor-inostat (SAHA, AdooQ) were dissolved in DMSO. Romidepsin was added at a final concentration of 100 nM and Vorinostat was added at final concentration of 10 μM for 24 h. p300 inhibitor A485 (Cayman) and C646 (Cayman) were dissolved in DMSO. A485 was added at a final concentration of 3 μM and C646 was added at a final concentration of 10 μM for 24 h for the gene expression analysis. For the HAT and HDAC double inhibition experiment, the keratinocytes were pre-treated with the HAT inhibitor for 6 h before the HDAC inhibitors were added for 24 h.

**Progenitor competition in epidermal regeneration**. The same number of kera-tinocytes transduced with GFP or DsRed were mixed and seeded on pieces of human dermis. GFP-expressing cells were treated with non-targeting shRNA virus and DsRed-expressing cells were treated with either non-targeting shRNA virus or NUP98/RAE1-targeting shRNA. Epidermal tissue was regenerated at the liquid-air interface for 6 days. Each piece of epidermal tissue was embedded in OCT, and was sectioned using Cyrostat. The tissue sections were fixed in 4% formaldehyde, and images were acquired using the EVOS FL Auto 2 microscope.

**Western blotting**. Western blot analysis was performed by loading 10–25 μg of cell lysate per lane for sodium dodecyl sulfate polyacrylamide gel electrophoresis. The gel was transferred to a nitrocellulose membrane and incubated with primary antibodies at 4 °C overnight. Secondary antibodies were incubated at room tem-perature for 1 h. The blots were imaged using Li-COR OdysseyCLx InfraRed Imaging System and images were analyzed using ImageStudio Lite software.

Antibodies used for western in this study include NUP98 C-7 (Santa Cruz Biotechnology, sc-74553), Lamin A/C (Santa Cruz Biotechnology, sc-376248), RAE1 (Santa Cruz Biotechnology, sc-393252), TPR (Bethyl, A300-828A), NUP153 (Bethyl A301-788A-M), DNMT1 (Cell Signaling, D63A6), and HDAC1 (Cell Signaling, D5C6U).

**qRT-PCR expression analysis**. For qRT-qPCR experiments, total RNA was extracted from keratinocytes using Quick-RNA MiniPrep Kit from Zymo Research, and cDNA was synthesized using SuperScript VILO cDNA synthesis kit. The qRT-qPCR analysis was performed using the QuantStudio3 QPCR system and samples were prepared using PowerUp SYBR Green Master Mix (Applied Biosystems). Samples were run in technical triplicates and normalized to the 18S control. Sta-tistical analysis was performed using GraphPad Prism8. All error bars present standard deviation of $2^{-(\Delta\Delta CT)}$ between biological triplicates. Sequences of the qPCR primers used in this study are included in Supplementary Data 7.

**RNA-seq**. RNA-seq libraries were prepared using NEBNext® Ultra™ II Directional RNA Library Prep Kit for Illumina. RNA-seq samples were sequenced by North-western sequencing core (NUseq) using Illumina HiSeq4000. The computational and bioinformatics pipelines were performed at Quest high performance com-puting facility at Northwestern University. The pipelines were constructed based on open-source software using nextflow framework. Briefly, Quality control using FastQC v0.11.9 and adapter trimming using Trimgalore v0.6.7 were performed on sequence reads from the RNA-Seq data. STAR v2.7.10a was used to align the reads to the reference genome (Hg38 from UCSC), and Salmon v1.5.2 was used for gene and transcripts quantification. Downstream analysis was performed in R (R. Foundation for Statistical Computing). R package DESeq2 was used for differential expression analysis[44]. Genes were filtered by average Log 2 (Fold Change) > 2, with individual Log 2 (Fold Change) > 1.5, and p value < 0.05.

**ChIP-seq**. Each ChIP sample was prepared using ~10 million keratinocytes crosslinked with 1% formaldehyde for 10 min at room temperature. The cross-linking was quenched using 0.125 M final of glycine for 5 min at room temperature. The crosslinked cells were then washed with PBS and nuclei extracted. Nuclei were lysed in lysis buffer at 40 μL per million cells. Lysate was sonicated until DNA was sheared to ~200 bp. Lysate was cleared by spinning at 15,000 × g for 15 min at 4 °C. Protein G beads (10 μL) were incubated at room temperature for 10 min with antibodies then 2 h at 4 °C before adding ChIP lysate overnight. Beads were washed and sample was eluted in elution buffer overnight at 67 °C. Eluted sample was treated with RNase A and Proteinase K. The DNA was purified using the DNA clean and concentrator kit (Zymo). For double-crosslinking, DSG (Sigma, 80424) was added to keratinocytes at the final concentration of 2 mM and incubated at room temperature for 30 min, before the addition of 1% formaldehyde for 10 min. ChIP DNA library samples were prepared using the NEBNext® Ultra™ II DNA Library Prep Kit for Illumina. The antibodies used for ChIP: NUP98 C-7 (Santa Cruz Biotechnology, sc-74553) was used for NUP98 ChIP experiments. HDAC1 (Diagenode, C15410325) was used for all HDAC1 ChIP experiments. The HA antibody(#66006, Proteintech) was used for all RAE1 ChIP-seq. ChIP-seq libraries were sequenced by NUseq using Illumina HiSeq4000. The quality of the raw sequencing files were assessed using FastQC software v0.11.9 and reads with low quality were trimmed using Trimgalore v0.6.4. The remaining ChIP-seq reads were mapped to hg38 (NCBI) using BWA-MEM v0.7.17 with default parameters. ChIP-seq peaks were called using MACS2 v2.2.7.1 with q value < 0.01 and narrow peak calling. Bigwig files were created using BEDTools v2.29.2 and scaled to 1 million mapped reads. Peak annotation was done using HOMER v4.11.

**Co-immunoprecipitation**. Undifferentiated keratinocytes were nuclei extracted (per IP). Cells were resuspended in 6 mL buffer A (10 mM Hepes pH 7.4, 1.5 mM MgCl2, 10 mM KCl, protease inhibitor (-) EDTA) and 6 mL buffer B (10 mM Hepes pH 7.4, 1.5 mM MgCl2, 10 mM KCl, 0.4% NP-40, protease inhibitor (-) EDTA). Cells were incubated on ice for 2 min and quickly pelleted. Cell resus-pension was repeated, and suspension was incubated on ice for 5 min. Cells were quickly pelleted and resuspended in 1 mL of nuclei lysis buffer (50 mM Tris pH 8, 0.05% NP-40, 10% glycerol, 2 mM MgCl2, 250 mM NaCl, protease inhibitor (-) EDTA). Nuclei were sheared using a 27^1/2-gauge needle 5 times. Lysed nuclei solution was incubated on ice for 30 min. Cell debris was pelleted at 4 C, 13,000 rpm for 10 min. 30 μL of protein G beads were washed with PBS (per IP). Comparable amount of NUP98 antibody C-7 (Santa Cruz Biotechnology, sc-74553), RAE1 antibody (Santa Cruz Biotechnology, sc-393252) or mouse IgG control (cell signaling) were incubated with washed beads for 10 min at room temp and 1 h at 4 °C. Antibody solution was removed from beads and 350 μL of nuclei lysate was added to the beads. Bead:lysate mixture was incubated overnight at 4 °C with rotation. For co-immunoprecipitation with RNase treatment, half of the nuclei lysate was treated with RNase A (final concentration 50 μg/mL) for 20 min at 30 °C. Lysate with and without RNase A treatment were added to beads overnight.

**Co-immunoprecipitation with crosslinking**. Undifferentiated keratinocytes were crosslinked using 1% formaldehyde for 10 min at room temperature. Crosslinking was quenched using 0.125 M glycine for 5 min at room temperature. Nuclei were

extracted using swelling buffer (0.1 M Tris pH7.6, 10 mM KOAc, 15 mM MgOAc, 1% NP-40, protease inhibitor) for 15 min on ice. Nuclei were pelleted and resuspended in nuclei lysis buffer (in PBS, 1% NP-40, 0.5% Sodium deoxycholate, 0.1% SDS, 10 mM EDTA, protease inhibitor). Nuclei were sheared using a $27^{1/2}$-gauge needle 10 times. Lysed nuclei solution was incubated on ice for 15 min. Lysate was sonicated for 4 rounds 30 s on 30 s off for 10 cycles. Cell debris was pelleted, and supernatant was added to protein G beads.

**Apoptosis assay.** Keratinocytes infected with control, NUP98, or RAE1 shRNA were seeded onto 24-well plate. A positive control was made by adding $H_2O_2$ (2 mM) to keratinocytes expressing control shRNA for 6 h at 37 °C. MitoView 633 (Biotium) and Aquaphile JC1 (Biotium) apoptosis assay kits were used to assess induction of apoptosis with loss of NUP98 or RAE1. JC1 or Mitoview was added at 1:1000 in keratinocyte culture medium, and incubated for 20 min in the $CO_2$ incubator. Hoechst was added at 10 μg/mL for 5 min to stain the DNA before image acquisition.

**Size-exclusion chromatography.** Thirty million undifferentiated keratinocytes were pelleted, and nuclei were extracted. Cells were resuspended in 250 μL of nuclei lysis buffer (50 mM Tris pH 8, 0.05% NP-40, 10% glycerol, 2 mM MgCl2, 250 mM NaCl, protease inhibitor (-) EDTA). Nuclei were sheared using a $27^{1/2}$-gauge needle 5 times. Lysed nuclei solution was incubated on ice for 30 min. Cell debris was pelleted at 4 °C, 13,000 rpm for 10 min. The supernatant was loaded onto a Superose 6 Increase 10/300 GL (29-0915-96 GE) using nuclei lysis buffer for running buffer at a flow rate of 0.3 mL/min. Five hundred microliters fractions were collected. Protein in the 500 μL fractions were precipitated using acetone.

**Clonogenicity assay.** Mitomycin C treated mouse fibroblasts 3T3 cells were used as a feeder layer in 6-well plates. One thousand keratinocytes were seeded onto the feeder layer and FAD media was changed every 2 days for 12 days. Cells were then washed with PBS to remove 3T3 cells and fixed in 1:1 acetone/methanol for 5 min. The plates were then air dried for 5 min and stained with crystal violet for visualizing the clones.

**Image quantification.** Nucleolar enrichment was quantified using the FIJI software. The fluorescent signals in the nucleus were measured using DAPI as a mask, and the fluorescent signals in the nucleolus were measured using fibrillarin as a mask. Nucleolar enrichment was then calculated by dividing the fluorescent signals in the nucleolus by the fluorescent signals in the nucleus.

**Statistics and reproducibility.** One-way ANOVA (non-parametric) with post-hoc test was performed for data analyses comparting more than two experimental groups The student t-test was used to analyze the statistical difference between two experimental groups. All these statistical analyses were performed using Prism. For qRT-PCR, all the experiments were performed at least 3 times. The average relative value was calculated based on technical triplicates were calculated, and statistical analyses were performed based on the average relative values from multiple independent experiments. For the relative expression quantification using western blotting, each experiment was replicated at least 3 times, and the relative levels were quantified using loading controls such as Lamin A/C. For shRNA-mediated knockdown, 3 independent shRNAs for NUP98 or RAE1 were validated in this study. For RNA-seq, two independent control shRNAs (non-targeting or GFP) were used in combination with three independent shRNAs for each target genes. All the ChIP-seq experiments were repeated at least 2 times. For inhibitor treatment, two independent inhibitors for each target were used to ensure reproducibility.

**Reporting summary.** Further information on research design is available in the Nature Portfolio Reporting Summary linked to this article.

## Data availability

All the RNA-seq and ChIP-seq data have been deposited to GEO, with the accession number #GSE150799. All other data are included in the Source Data file (Supplementary Data 8). All the unprocessed scans for the western blots are also included in Supplementary Fig. 9. Questions about the data should be addressed to corresponding author.

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

## Acknowledgements

We thank Drs. Jason Brickner, Curt Horvath, and Yue Yang at Northwestern University for discussions about this project. We thank the core facilities, including Skin Biology & Diseases Resource-Based Center (SBDRC, human skin specimen acquisition) and RNA-seq and ChIP-seq sample sequencing performed NUseq core at Northwestern University. This work is supported by the Searle Leadership Fund, the NIH/NIAMS R01 (AR075015), the American Cancer Society Research Scholar Grant (RSG-21-018-01-DDC), and a Pilot & Feasibility (P&F) Award from the Skin Biology & Diseases Resource-Based Center (SBDRC, AR057216) at Northwestern University to X.B.

## Author contributions

X.B. and A.E.N. designed the project and the experiments and wrote the manuscript; A.E.N., L.A.B., P.J.H., S.M.L., J.K., Z.R., X.B. performed the experiments and/or analyzed the data; X.B. obtained funding for supporting this project.

## Competing interests

The authors declare no competing interests.
