## [Peer Review File · Communications Biology]

Reviewers' comments:

Reviewer #1 (Remarks to the Author):

The paper by Neely et al. provides a set of observations that link NUP98 and RAE1 and are suggestive of functions in progenitor maintenance for NUP98 away from NPCs. The data within the manuscript are fairly presented overall and expected to be of interest to others in the scientific community. The model that emerges from the work is an interesting one involving regulated localization of a NUP98-RAE1 complex via histone modifications. However, I do not feel the data presented supports all aspects of this model, especially related to RAE1, as noted below.

Major points:

1. The data well establishes a connection between NUP98 and HDAC1 at transcription sites and NUP98 and RAE1 in terms of nucleolar localization, plus a connection between all three at the level of gene expression regulation with DNMT1. However, the paper title, discussion, and figure model all suggest that RAE1 is working at transcription sites with NUP98, but there is no data to support this. Is RAE1 with NUP98 at transcription sites? Or could RAE1 act up or downstream? Mapping RAE1 binding sites using ChIP-seq is needed to support the model as presented.
2. Figures 1b-d are interpreted to suggest NUP98 and RAE1 form a unique complex away from NPCs, but it is not clear why NUP98 does not also pull down TPR as a component of NPCs in figure 1c. Due to complex stability, maybe TPR is the component lost from NPCs under the IP conditions used? To address this, and other possibilities, blotting for other Nups is needed. For example, multiple Nups could be assessed using the mAb414 antibody after IP with NUP98 or RAE1 antibodies. An IP of intact NPCs using mAb414 or other Nup antibodies to show NUP98 association, and not RAE1, would lend further evidence to the existence of a soluble NUP98-RAE1 complex that is independent of Nup98 bound NPCs. Given the role of RAE1 in mRNA export, and links provided here between NUP98-RAE1 and gene expression, it would also be valuable to know if RNA is a component of this complex? The authors should try the IPs in the context of an RNase treatment to address this point.

Other points to address:

3. The significance of nucleolar localization of NUP98 and RAE1 under HDAC inhibition is unclear. Is it a global loss of binding sites under this non-physiological drug condition that leads to the proteins mis-localizing? Or are there physiological conditions where changes in HDAC activity lead to regulated changes in NUP98 and RAE1 localization? If such conditions are known, they should be tested to provide further significance to the observation of nucleolar localization.
4. The observation that KD of NUP98 and RAE1 lead to DNMT1 level changes provides a mechanism to explain some of the gene expression alterations and mechanism by which progenitors are maintained (as presented in the model figure). I am not sure why these data are in the supplementary data and would suggest putting this in a main figure.
5. The authors introduction suggests nothing is known about NUP98 recruitment to chromatin, which fails to address previous work that suggested DEAD-box proteins may play a role (see: <https://pubmed.ncbi.nlm.nih.gov/28221134/>). The text states "Without any DNA or histone binding domains, how NUP98 binds to specific genomic sites also remains unknown". It should also be noted that the above paper also identified an interaction between RAE1 and NUP98. The text should be edited to incorporate this information in the intro, results, and discussion sections.
6. On page 7, the authors note "NUP98 has been observed to bind to different genomic regions in the context of different cell types.", but no references are provided. This lack of references occurs in other places too. Another example is "Although p300 was identified as an interacting protein with NUP98-

fusion proteins in hematopoietic malignancies, we found...". The authors should carefully review their citations and add additional references where needed.

7. On page 7, the authors state "The majority of these genes also show similar upregulation or downregulation with RAE1 knockdown.", but it is unclear what the majority is or what this data is.

8. The authors state "An average person turns over about 30 trillion cells in a single day, from a spectrum of self-renewing somatic tissues such as blood, gut epithelium, and skin epidermis1." I would check this statement as the cited paper states this is ~330 billion cells.

9. In terms of writing and data presentation, some paragraphs are rather brief (see page 9 and 10) and presented as small stand-alone fragments. As a reader, I would appreciate a bit more explanation and incorporation of these sections into a larger framework/model. This would help the general reader follow and appreciate the findings.

Reviewer #2 (Remarks to the Author):

Neely et al. reported on the involvement of Rae1-Nup98 in control epigenetic regulators, escaping from nucleolar aggregation. The authors also delineated the pathway that sustains progenitor function through HDAC-dependent chromatin targeting to escape from nucleolar localization. Using NUP98 or RAE1 knockdown, they performed transcriptome profiling using RNA-seq experiments. The authors further discovered NUP98's genomic targeting was diminished upon HDAC inhibition, which also dysregulated Rae1 and Nup98's target genes. The manuscript showed (mechanistically), how Rae1 and NUP98 functions upstream to key epigenetic regulators in epidermal progenitor maintenance. This is another very exciting result that shows how NPC involves in the regulation of epidermal differentiation. Therefore, I recommend the acceptance of this manuscript for publication.

General comment:

Wherever this study will be published, the authors should be very cautious to properly acknowledge other teams who previously made the critical discoveries to clarify what was already known Rae1-Nup98-HDAC1 (That might be how editors searched the relevant/suitable potential reviewers).

- ...different nucleoporins participate in the regulation of epidermal ...differentiation... PMID: 29217659
- Rae1-nup98-HDAC1 was reported in PMID: 21467841,
- Nup98-HDAC1 was reported in PMID: 16651408

Reviewer #3 (Remarks to the Author):

In the manuscript by Neely et al, the Authors show that a complex formed by NUP98 and RAE1 is highly expressed in epidermal progenitors in the nucleoplasm while their expression is decreased during differentiation. Silencing of both NUP98 and RAE1 determined the reduction of progenitor proliferation and regenerative capacity. By ChIPseq the Authors also found that NUP98 and RAE1 bind to genes involved in chromatin regulation, transcription and modification. RT-PCR data confirmed that chromatin remodeling enzymes such as DNMT1, EZH2, UHRF1 were NUP98 and RAE1 targets. Further, NUP98 peaks co-localized with both activatory and inhibitory histone marks along with Pol2 and HDAC1. The interaction of NUP98 and RAE1 with HDAC1 was confirmed by co-immunoprecipitation and the role of HDACs was also investigated by experiments of ChIPseq using HDAC inhibitors for class I. Inhibition of HDACs resulted in an inter-dependent localization of NUP98 and RAE1 in the nucleolus.

Overall this work is well done and of interest providing new insights in the role exerted by NUP98 and RAE1 in the control of epigenetic mechanisms in keratinocyte progenitors. Below some comments to improve the manuscript.

Specific comments:

Statistics: In all the figures (e.g. fig.1 e,f; Fig.2 d,f,h; Fig. 3 n,o; Fig.4 g; Fig.5 e,g; Fig.6 b,d) a T test is applied for the comparison of more than two samples. This is inappropriate and the comparison of multiple samples should be performed with ANOVA (or non-parametric test for non-normally distributed samples) followed by a post-hoc test. Further, a statistic paragraph is completely missing in the method section.

Figure S1 and S3 e,f lack densitometry and statistics.

It would be better to characterize isolated proliferating or differentiated keratinocytes with progenitor and differentiation markers to assess purity and identity of the population studied.

From the clonogenicity assay it seems that the cells silenced for NUP98/RAE1 completely stop proliferating. Do NUP98 and RAE1 silencing induce also apoptosis?

Treatment with HDAC inhibitors induces NUP98 and RAE1 detachment from chromatin and accumulation in the nucleolus. Are NUP98 and RAE1 targets of acetylation? It is possible that increased levels of acetylated NUP98 or RAE1, following HDACi, influence their localization within the nucleus? Does HDAC1 have a role in activation/repression of NUP98 target genes or affect NUP 98 binding to chromatin? Does the specific silencing of HDAC1 have the same effect of HDACi on NUP98 binding on target genes?

Introduction and discussion could be improved adding more information on nucleoporins and stem cell regulation (introduction) and on the relationship between Nups and HDACs in the regulation of Nup post-translational modifications (e.g acetylation) and in HDAC-dependent Nup chromatin positioning by interaction with HAT/HDAC.

The DNMT1 antibody is not indicated in the method section.

Point-by-Point response to reviewers

(original comments in gray, responses in black, new data in figures in Blue.)

REVIEWER #1 (REMARKS TO THE AUTHOR):

The paper by Neely et al. provides a set of observations that link NUP98 and RAE1 and are suggestive of functions in progenitor maintenance for NUP98 away from NPCs. The data within the manuscript are fairly presented overall and expected to be of interest to others in the scientific community. The model that emerges from the work is an interesting one involving regulated localization of a NUP98-RAE1 complex via histone modifications. However, I do not feel the data presented supports all aspects of this model, especially related to RAE1, as noted below.

We appreciate the positive comments that the data “are fairly presented overall” and “expected to be of interest to others”. We are grateful for the reviewer’s constructive suggestions to further strength the model. We have incorporated additional data, including the RAE1 ChIP-seq experiment. The details are listed below.

Major points:

1. The data well establishes a connection between NUP98 and HDAC1 at transcription sites and NUP98 and RAE1 in terms of nucleolar localization, plus a connection between all three at the level of gene expression regulation with DNMT1. However, the paper title, discussion, and figure model all suggest that RAE1 is working at transcription sites with NUP98, but there is no data to support this. Is RAE1 with NUP98 at transcription sites? Or could RAE1 act up or downstream? Mapping RAE1 binding sites using ChIP-seq is needed to support the model as presented.

We have now incorporated RAE1 ChIP-seq data to the paper. We had tested multiple commercially available RAE1 antibodies, which did not result in high-quality ChIP-seq data. To overcome this technical barrier, we generated an HA-tagged RAE1 construct and expressed it in keratinocytes. We confirmed that this HA-tag does not interfere with RAE1’s association with NUP98 using co-immunoprecipitation. Using an HA antibody, we successfully generated RAE1 ChIP-seq using the double crosslinking condition (DSG & formaldehyde). The new data are now included as Fig.S3. We found that RAE1 is enriched in the majority (83%) of NUP98 ChIP-seq peaks, and these shared regions are mainly located within 3 kb of the nearest TSS. 96 out of the 101 NUP98 direct target genes also have RAE1 binding, and these include DNMT1, UHRF1 and EZH2. These data support the model that NUP98 and RAE1 bind together near the transcription start sites in gene regulation.

2. Figures 1b-d are interpreted to suggest NUP98 and RAE1 form a unique complex away from NPCs, but it is not clear why NUP98 does not also pull down TPR as a component of NPCs in figure 1c. Due to complex stability, maybe TPR is the component lost from NPCs under the IP conditions used? To address this, and other possibilities, blotting for other Nups is needed. For example, multiple Nups could be assessed using the mAb414 antibody after IP with NUP98 or RAE1 antibodies. An IP of intact NPCs using mAb414 or other Nup antibodies to show NUP98 association, and not RAE1,

would lend further evidence to the existence of a soluble NUP98-RAE1 complex that is independent of Nup98 bound NPCs. Given the role of RAE1 in mRNA export, and links provided here between NUP98-RAE1 and gene expression, it would also be valuable to know if RNA is a component of this complex? The authors should try the IPs in the context of an RNase treatment to address this point.

We thank the reviewer for the suggestions to further clarify the NUP98-RAE1 interaction. The immunoprecipitation experiment and the size-exclusion chromatography experiment were performed using nuclear extract prepared using the same “nuclear lysis buffer” (50 mM Tris pH 8, 0.05% NP-40, 10% glycerol, 2 mM MgCl₂, 250 mM NaCl, protease inhibitor). This buffer contains very low detergent, and it had been optimized for characterizing protein-protein interactions in the soluble fractions of the nuclei. We apologize for the confusion, as the intention of this experiment was not to purify the intact nuclear pore complexes, which would require a different buffer. Under this specific condition, both the size-exclusion chromatography and the immunoprecipitation experiment indicate that NUP98 and RAE1 form a separate sub complex away from other NUPs such as TPR.

We further performed the co-immunoprecipitation suggested by the reviewer. Immunoprecipitated proteins by the NUP98 or RAE1 antibody was probed by the mAb414 antibody. While mAb414 recognizes multiple bands from the input lysate as expected, this banding pattern was not captured from NUP98 or RAE1 IP (new data included as Fig. S1b), suggesting that NUP98 or RAE1 does not associate with the NPC under this immunoprecipitation condition. We also performed NUP98 IP with or without RNase treatment, and we found that the NUP98-RAE1 interaction was not disrupted by RNase (new data included in Fig. S1a). These data suggest that the NUP98-RAE1 interaction does not require RNA. This direct protein-protein interaction between NUP98 and RAE1 agrees with a previously published paper (Ren et al., PNAS, 2010), which presents a crystal structure of human RAE1 in complex with the Gle2-binding sequence of NUP98. These data suggest that the interaction between NUP98 and RAE1 can occur independent from the nuclear pores. This interaction is further supported by the ChIP-seq data as well as the re-targeting of both proteins to the nucleolus.

Other points to address:

3. The significance of nucleolar localization of NUP98 and RAE1 under HDAC inhibition is unclear. Is it a global loss of binding sites under this non-physiological drug condition that leads to the proteins mis-localizing? Or are there physiological conditions where changes in HDAC activity lead to regulated changes in NUP98 and RAE1 localization? If such conditions are known, they should be tested to provide further significance to the observation of nucleolar localization.

We appreciate this great question, and we have looked further into this direction. First, we found that HDAC protein expression is slightly reduced in keratinocyte differentiation (new data included as Fig. S5c). We further examined HDAC1 chromatin binding using ChIP-seq, comparing the progenitor state and the differentiation state (new data included as Fig. S5d-h). In agreement with the western blots, HDAC1 ChIP-

seq enrichment was only mildly reduced in differentiation. On the other hand, NUP98 protein expression and NUP98 Chromatin binding is strongly reduced in differentiation. These data suggest that HDAC facilitates NUP98 chromatin binding only in the progenitor state, where NUP98's high expression allows it to bind chromatin in addition to its nuclear pore localization.

This nucleolar enrichment of NUP98 was previously observed in cells treated with actinomycin-D (Oka et al., MBoC, 2010). In this context, another NUP98-interacting-protein CRM1 also enriches to the nucleolus together with NUP98. Furthermore, CRM1 inhibition by its inhibitor LMB was sufficient to block the nucleolar enrichment of both NUP98 and CRM1 in this context. Another study further showed that knockdown of the RNA polymerase I mimicked the actinomycin-D treatment in inducing CRM1's nucleolar localization, which is also dependent the ribosome-export receptor NMD3 (Bai et al., Nucleus, 2013). Furthermore, ribosomal transcription is regulated by acetylation, including the roles of HDAC (Pelletier et al., Molecular Cell, 2000). In our system, we found that HDAC inhibition also induced the nucleolar localization of CRM1, which is also sensitive to LMB treatment (Fig. R1). Thus, our findings of NUP98 and RAE1's nucleolar enrichment is likely to be involved in ribosomal transcription or transport, which is not directly connected to the focus of this paper but will be a very interesting direction for future investigation. We have also expanded the discussion section to include these comments.

4. The observation that KD of NUP98 and RAE1 lead to DNMT1 level changes provides a mechanism to explain some of the gene expression alterations and mechanism by which progenitors are maintained (as presented in the model figure). I am not sure why these data are in the supplementary data and would suggest putting this in a main figure.

We thank the reviewer for this suggestion. This set of western blots together with quantification are now moved to the main figures (Fig. 3p-s).

5. The authors introduction suggests nothing is known about NUP98 recruitment to chromatin, which fails to address previous work that suggested DEAD-box proteins may play a role (see: <https://pubmed.ncbi.nlm.nih.gov/28221134/>). The text states "Without any DNA or histone binding domains, how NUP98 binds to specific genomic sites also remains unknown". It should also be noted that the above paper also identified an interaction between RAE1 and NUP98. The text should be edited to incorporate this information in the intro, results, and discussion sections.

We appreciate the suggestion. We have revisited this very interesting DHX9 paper pointed out by the reviewer. Fig.4 of this DHX9 paper shows that DHX9 knockdown does not alter NUP98 localization inside the cell. Fig. 5 of this DHX9 paper shows that GFP-NUP98 overexpression recruits DHX9 to intranuclear foci. These data indicate that NUP98 modulates DHX9 subcellular localization, but not vice versa. This paper also suggests that the interaction between NUP98 and DHX9 involves RNA, which appears to be different from the direct protein-protein interaction between NUP98 and RAE1. Therefore, we have tuned down the statement pointed out by the reviewer in the introduction section, changed it from “unknown” to “largely unclear”. We have further incorporated this DHX9 paper to the results and discussion sections.

6. On page 7, the authors note “NUP98 has been observed to bind to different genomic regions in the context of different cell types.”, but no references are provided. This lack of references occurs in other places too. Another example is “Although p300 was identified as an interacting protein with NUP98-fusion proteins in hematopoietic malignancies, we found...”. The authors should carefully review their citations and add additional references where needed.

We thank the reviewer for pointing this out. We have added these citations and other additional citations to the paper.

7. On page 7, the authors state “The majority of these genes also show similar upregulation or downregulation with RAE1 knockdown.”, but it is unclear what the majority is or what this data is.

We apologize for the confusion, and we have edited the text to clarify that “the majority” refers to “the NUP98 101 direct target genes”.

8. The authors state “An average person turns over about 30 trillion cells in a single day, from a spectrum of selfrenewing somatic tissues such as blood, gut epithelium, and skin epidermis1.” I would check this statement as the cited paper states this is ~330 billion cells.

We appreciate that the reviewer pointed this out. We have corrected this number in the text.

9. In terms of writing and data presentation, some paragraphs are are rather brief (see page 9 and 10) and presented as small stand-alone fragments. As a reader, I would appreciate a bit more explanation and incorporation of these sections into a larger framework/model. This would help the general reader follow and appreciate the findings.

We thank the reviewer for pointing this out. We have incorporated them to one section, and edited the writing to fit them to a larger framework.

REVIEWER #2 (REMARKS TO THE AUTHOR):

Neely et al. reported on the involvement of Rae1-Nup98 in control epigenetic regulators, escaping from nucleolar aggregation.

The authors also delineated the pathway that sustaining progenitor function through HDAC-dependent chromatin targeting to escape from nucleolar localization. Using NUP98 or RAE1 knockdown, they performed transcriptome profiling using RNA-seq experiments. The authors further discovered NUP98's genomic targeting was diminished upon HDAC inhibition, which also dysregulated Rae1 and Nup98's target genes. The manuscript showed (mechanistically), how Rae1 and NUP98 functions upstream to key epigenetic regulators in epidermal progenitor maintenance. This is another very exciting results that showing how NPC involves in the regulation of epidermal differentiation. Therefore, I recommend the acceptance of this manuscript for publication.

We appreciate the positive comments from this reviewer.

General comment:

Wherever this study will be published, the authors should be very cautious to properly acknowledge other teams who previously made the critical discoveries to clarify what was already known Rae1-Nup98-HDAC1 (That might be how editors searched the relevant/suitable potential reviewers).

- ...different nucleoporins participate in the regulation of epidermal ...differentiation...

PMID: 29217659

- Rae1-nup98-HDAC1 was reported in PMID: 21467841,

- Nup98-HDAC1 was reported in PMID: 16651408

We thank the reviewer for pointing out these relevant papers. These are now cited and incorporated to the manuscript.

Reviewer #3 (Remarks To The Author):

In the manuscript by Neely et al, the Authors show that a complex formed by NUP98 and RAE1 is highly expressed in epidermal progenitors in the nucleoplasm while their expression is decreased during differentiation. Silencing of both NUP98 and RAE1 determined the reduction of progenitor proliferation and regenerative capacity. By ChIPseq the Authors also found that NUP98 and RAE1 bind to genes involved in chromatin regulation, transcription and modification. RT-PCR data confirmed that chromatin remodeling enzymes such as DNMT1, EZH2, UHRF1 were NUP98 and RAE1 targets. Further, NUP98 peaks co-localized with both activatory and inhibitory histone marks along with Pol2 and HDAC1. The interaction of NUP98 and RAE1 with HDAC1 was confirmed by co-immunoprecipitation and the role of HDACs was also investigated by experiments of ChIPseq using HDAC inhibitors for class I. Inhibition of HDACs resulted in an inter-dependent localization of NUP98 and RAE1 in the

nucleolus.

Overall this work is well done and of interest providing new insights in the role exerted by NUP98 and RAE1 in the control of epigenetic mechanisms in keratinocyte progenitors. Below some comments to improve the manuscript.

We appreciate the positive comments from the reviewer, and we have made the changes as suggested by the reviewer to improve the manuscript.

Specific comments:

Statistics: In all the figures (e.g. fig.1 e,f; Fig.2 d,f,h; Fig. 3 n,o; Fig.4 g; Fig.5 e,g; Fig.6 b,d) a T test is applied for the comparison of more than two samples. This is inappropriate and the comparison of multiple samples should be performed with ANOVA (or non-parametric test for non-normally distributed samples) followed by a post-hoc test. Further, a statistic paragraph is completely missing in the method section.

We thank the reviewer for the suggestion. The statistical analyses for these figures have now been updated, by using have added the appropriate statistics to all figures in addition to adding a section in the methods section describing out statistical analysis.

Figure S1 and S3 e,f lack densitometry and statistics.

We have now included quantification and statistics to the figures pointed out by the reviewer.

It would be better to characterize isolated proliferating or differentiated keratinocytes with progenitor and differentiation markers to assess purity and identity of the population studied.

We apologize for the confusion. The differentiated keratinocytes were not directly isolated from skin. Instead, these cells were induced to differentiation from the progenitor-state keratinocytes using high fluency and high calcium. This method is widely used in the field and the cells show distinct morphology (Fig. R1). To further characterize these cells and to validate the findings from the literature, we have now included western blots and quantification, showing the reduction of DNMT1 as keratinocytes switch from the progenitor state to the differentiation state (new data included as Fig. S4 a-b).

Fig. R2. Representative Images of primary human keratinocytes cultured in the progenitor state or the differentiation state.

From the clonogenicity assay it seems that the cells silenced for NUP98/RAE1 completely stop proliferating. Do NUP98 and RAE1 silencing induce also apoptosis?

We have now included data showing NUP98 or RAE1 knockdown does not induce apoptosis, using two different apoptosis indicators Mitoview and JC-1 (new data included as Fig. S2 f,g).

Treatment with HDAC inhibitors induces NUP98 and RAE1 detachment from chromatin and accumulation in the nucleolus. Are NUP98 and RAE1 targets of acetylation? It is

possible that increased levels of acetylated NUP98 or RAE1, following HDACi, influence their localization within the nucleus? Does HDAC1 have a role in activation/repression of NUP98 target genes or affect NUP 98 binding to chromatin? Does the specific silencing of HDAC1 have the same effect of HDACi on NUP98 binding on target genes?

We appreciate this great question from the reviewer. To determine if acetylation is involved, we treated keratinocytes with p300 inhibitors in combination with HDAC inhibition. Strikingly, p300 inhibition blocked the nucleolar targeting of NUP98/RAE1 induced by HDACi (new data included as Fig. 6). According to the PhosphoSitePlus database, acetylation sites on NUP98 and RAE1 were detected using mass spectrometry. Thus, it is very likely that acetylation is involved in modulating NUP98/RAE1's nucleolar localization.

To determine if HDAC1 has a role in regulating NUP98 direct targets, we knocked down HDAC1 using three independent shRNA and performed qPCR on NUP98 direct targets. All three HDAC1 shRNAs consistently showed significant reduction of the direct target genes such as DNMT1, EZH2 and UHRF1 (new data included as Figure S5 b), suggesting that HDAC1 is involved in regulating the expression of NUP98 direct target genes.

We further investigated if HDAC1 knockdown is sufficient to drive NUP98's nucleolar enrichment, but did not observe a strong enrichment. It was possible that the knockdown strategy was not sufficient to deplete all the HDAC, and the residual HDAC could still be sufficient to alleviate the nucleolar localization. It is also possible that other HDACs could be involved inducing the nucleolar enrichment of NUP98/RAE1.

Introduction and discussion could be improved adding more information on nucleoporins and stem cell regulation (introduction) and on the relationship between Nups and HDACs in the regulation of Nup post-translational modifications (e.g acetylation) and in HDAC-dependent Nup chromatin positioning by interaction with HAT/HDAC.

We thank the reviewer for this suggestion. The introduction section is now updated to incorporate more information and citations on the roles of NUPs in stem cell regulation. We have also expanded and improved the discussion section to incorporate the roles of protein acetylation in the subnuclear location of NUP98 and RAE1.

The DNMT1 antibody is not indicated in the method section.

We apologize for missing this information in the previous submission. The specific DNMT1 antibody used in this study is now included in the methods section.

REVIEWERS' COMMENTS:

Reviewer #1 (Remarks to the Author):

In this revision, the authors have addressed the questions and concerns raised previously, which has significantly strengthened the manuscript.